# Bridging Structure and Semantics: Uncertainty-Modulated Dual-Path Diffusion for Robust Text-Attributed Graph Learning

Zhizhi Yu [1]  Jiachen Liu [1]  Qingyu Li [2]  Dongxiao He [1]  Di Jin [1]

## Abstract

Representation learning on text-attributed graphs (TAGs) is crucial for real-world applications, as it enables effective modeling of both rich node semantics and complex graph structure. Nevertheless, this task is intrinsically challenging due to structural–semantic mismatch stemming from divergent modality distributions, as well as dual-source noise inherent in node textual content and graph structure. Existing approaches often enforce a rigid fusion of distinct modalities while overlooking their inherent noise, which inevitably results in persistent distribution gaps and amplifies mixed interference during information propagation. To address these issues, we propose UDPD, an Uncertainty-modulated Dual-Path Diffusion model for robust text-attributed graph learning. Specifically, we first employ a dual-perspective node encoding strategy to separately learn semantic and structural embeddings. We then introduce a cooperative diffusion paradigm with parallel semantic and structural branches, where mutual guidance enables progressive alignment of different distributions while effectively suppressing modality inherent noise. Crucially, the reverse process is guided by node uncertainty, which is used to adaptively modulate cross-branch interaction strength, ensuring robust coupling and maximizing denoising effectiveness. Extensive experiments on five public benchmarks demonstrate the effectiveness and superiority of our UDPD over state-of-the-art baselines. Our code is available at https://github.com/hedongxiao-tju/UDPD.

[1]College of Intelligence and Computing, Tianjin University, Tianjin, China [2]College of Computer Science (College of Software), College of Artificial Intelligence, Inner Mongolia University, Hohhot, China. Correspondence to: Dongxiao He <hedongxiao@tju.edu.cn>.

*Proceedings of the 43rd International Conference on Machine Learning*, Seoul, South Korea. PMLR 306, 2026. Copyright 2026 by the author(s).

## 1. Introduction

Text-attributed graphs (TAGs), where nodes are enriched with textual content and interconnected through meaningful topological relations (Yang et al., 2021; Wang et al., 2025b), have become a prevalent data structure in the era of big data, serving as a cornerstone for modeling sophisticated real-world systems. As evidenced across diverse domains, TAGs are prevalent in academic networks, e-commerce platforms, as well as social media (Hu et al., 2020). Unlike plain graphs that focus exclusively on connectivity, representation learning on TAGs enables models to capture both the rich node semantics and complex graph structure (Yang et al., 2015). On this basis, jointly leveraging these complementary modalities is crucial for uncovering the underlying graph patterns and facilitating a range of downstream tasks (Zhao et al., 2023; Bo et al., 2025).

In recent years, Graph Neural Networks (GNNs), a class of neural networks tailored for graph-structured data, have demonstrated remarkable success in capturing node representations, and have been widely applied in tackling graph analytical tasks (Kipf & Welling, 2017). In the context of TAGs, mainstream approaches typically derive node representations through the propagation and aggregation of attributes along graph topology (Wu et al., 2021; Yan et al., 2023). Despite their success, such methods expose a key vulnerability: they heavily rely on the assumption that the graph topology is well aligned with node semantics (Zhu et al., 2020). When this assumption fails, the message passing mechanism may become counterproductive, which inadvertently propagates noise and amplifies interference, significantly degrading the quality of learned node representations (Zhang & Zitnik, 2020; Zhou et al., 2025a).

This assumption, however, is rarely satisfied in real-world TAGs due to inherent characteristics of the data. In particular, **structural–semantic mismatch** naturally arises from the heterogeneous modality distributions of graph topology and node textual content (Wang et al., 2025a). Graph topology often reflects community structures or functional links, whereas node textual content encodes node-specific semantics, leading to substantial discrepancies between their respective feature spaces. In practice, we observe that structural-semantic mismatch typically manifests in four

distinct scenarios: cross-field connections (different topics), varying focuses within the same topic, mismatch in specialized domains, and direct contradictions. Quantitative analysis reveals that the semantic similarity between neighboring nodes is often surprisingly low, with high Jensen-Shannon Divergence (JSD) between structural and semantic distributions. Detailed statistics and case discussions are provided in Appendix B. Not only that, TAGs are also inevitably affected by **dual-source noise** from both graph topology and node textual content (Linkerhägner et al., 2025). Spurious or task-irrelevant edges introduce structural noise, while redundant or irrelevant textual information further obscures informative semantic signals, jointly hindering robust representation learning. Unfortunately, most GNN variants fall short of effectively addressing these intrinsic conflicts (Zhou et al., 2025b; Wang et al., 2025c). Instead, they perform neighborhood propagation and aggregation directly on raw inputs, without explicitly accounting for distributional discrepancies or actively suppressing noise, limiting robustness in practical scenarios.

Drawing inspiration from recent advancements in Diffusion Models (DMs), we explore their powerful generative capabilities for purifying graph data (Ho et al., 2020). Yet, directly applying DMs to TAGs faces two critical hurdles. First, unlike vision data characterized by continuous and regular signals, TAGs typically rely on raw node attributes that are discrete and sparse, for which naive diffusion processes are ill-suited and may even degrade representation quality (Austin et al., 2021). Second, independently denoising graph topology and node semantics overlooks their intrinsic coupling (Jo et al., 2022). Uncoordinated diffusion processes tend to drive the two modalities toward divergent distributions, thereby amplifying structural–semantic misalignment (Vignac et al., 2023). As a result, the key difficulty lies in establishing a unified diffusion mechanism that can jointly purify both structural and semantic information while facilitating their mutual alignment.

To address the aforementioned issues, we propose a novel **U**ncertainty-modulated **D**ual-**P**ath **D**iffusion model for robust text-attributed graph learning, termed UDPD. Inspired by DMs, we reformulate node representation learning as a "forward-reverse" process. Specifically, we first utilize a dual-perspective encoding strategy to independently derive semantic and structural embeddings. On this basis, we introduce a cooperative diffusion paradigm where structural and semantic branches operate in parallel. Instead of independent diffusion, these two branches provide mutual conditioning to each other, which progressively aligns their distributions and filters out noise. Furthermore, we incorporate an uncertainty-modulated condition denoising mechanism to quantify node uncertainty and adaptively regulate the interaction intensity, thereby ensuring robust cross-branch coupling and maximizing the denoising efficacy.

We make the following main contributions:

- We find that most existing TAG representation learning methods overly focus on passive information propagation, while largely overlooking the inherent structural-semantic mismatch and dual-source noise, resulting in amplified interference and degraded representations.
- We propose UDPD, a cooperative dual-path diffusion approach that aligns structural and semantic distributions via mutual guidance, together with an uncertainty-modulated conditional mechanism that adaptively controls cross-branch interactions for robust denoising.
- Extensive experiments on five public benchmark datasets demonstrate that our proposed UDPD significantly outperforms state-of-the-art baselines, validating its rationality and effectiveness.

## 2. Preliminary

We first present notation and problem formulation, then introduce graph neural networks and diffusion models.

### 2.1. Notation and Problem Formulation

Given a text-attributed graph $\mathcal{G} = (\mathcal{V}, \mathcal{E}, \mathcal{T})$, where $\mathcal{V} = \{v_1, \ldots, v_n\}$ denotes the set of $n$ nodes, and $\mathcal{E}$ represents the set of $m$ edges connecting these nodes. For any edge $(v_i, v_j) \in \mathcal{E}$, nodes $v_i$ and $v_j$ are considered neighbors. We denote the set of one-hop neighbors of node $v_i$ as $N(v_i) = \{v_j \mid (v_i, v_j) \in \mathcal{E}\}$, and $N^{(l)}(v_i)$ represents the set of $l$-hop neighbors of node $v_i$.

In a TAG, each node $v_i$ is associated with a raw text description $t_i \in \mathcal{T}$, which encapsulates rich semantic information. Furthermore, let $\mathcal{Y} = \{y_1, \ldots, y_C\}$ denote the set of ground-truth labels, where $C$ represents the total number of classes. The primary objective is to learn effective node representations by leveraging both the graph structure and node semantics for downstream tasks.

### 2.2. Graph Neural Networks

For graph-structured data, Graph Neural Networks (GNNs) (Klicpera et al., 2019; Graziani et al., 2024) typically adopt a message passing mechanism to acquire node embeddings. Specifically, each node first aggregates its neighbors' embeddings, and then combines them with its own embedding to update itself, represented as:

$$m_i^{(l)} = \text{AGGREGATE}^{(l)} \left( \left\{ h_j^{(l-1)} \mid v_j \in N(v_i) \right\} \right),$$
$$h_i^{(l)} = \text{UPDATE}^{(l)} \left( h_i^{(l-1)}, m_i^{(l)} \right), \quad (1)$$

where $h_i^{(l)}$ denotes the embedding of node $v_i$ in $l$-th layer, and $h_i^{(0)}$ is the initial node embedding obtained by encoding raw text description $t_i$. AGGREGATE collects information

from neighbors via operations such as summation, mean pooling, or weighted averaging, while UPDATE merges the ego-information with the aggregated message through addition or concatenation.

## 2.3. Diffusion Models

Diffusion Models (DMs) (Ho et al., 2020; Li et al., 2025a; Li & Yang, 2025) typically consist of two fundamental processes: the forward process and the reverse process.

**Forward Process.** Given an initial data sample $x_0$, the forward process progressively injects Gaussian noise via a fixed Markov chain over $T$ steps. Formally, the transition probability from state $x_{t-1}$ to $x_t$ is defined as:

$$q(x_t|x_{t-1}) = \mathcal{N}(x_t; \sqrt{1-\beta_t}x_{t-1}, \beta_t I), \quad (2)$$

where $t \in \{1, \ldots, T\}$ denotes the diffusion step, $\beta_t$ represents the pre-defined variance schedule at each step, $I$ is the identity matrix, and $\mathcal{N}$ is the Gaussian distribution used to sample. For computational convenience, we define $\alpha_t = 1 - \beta_t$ and the cumulative product $\bar{\alpha}_t = \prod_{s=1}^{t} \alpha_s$.

**Reverse Process.** The reverse process aims to recover the original data distribution from Gaussian noise by learning a parameterized distribution $p_\theta$ that approximates the true reverse transition $x_t \rightarrow x_{t-1}$:

$$p_\theta(x_{t-1}|x_t) = \mathcal{N}(x_{t-1}; \mu_\theta(x_t, t), \Sigma_\theta(x_t, t)), \quad (3)$$

where $\mu_\theta$ and $\Sigma_\theta$ denote the mean and covariance parameterized by a neural network, respectively.

**Training and Inference.** DMs employ distinct denoising strategies during training and inference phases:

- **During training**, a **single-step reconstruction** strategy is adopted. Mathematically, given the predicted noise $\hat{\epsilon}_\theta(x_t, t)$, we directly estimate the initial state $\hat{x}_0$ from the current noisy state $x_t$:

$$\hat{x}_0 = \frac{1}{\sqrt{\bar{\alpha}_t}} \left(x_t - \sqrt{1-\bar{\alpha}_t} \cdot \hat{\epsilon}_\theta(x_t, t)\right). \quad (4)$$

- **During inference**, a **step-wise denoising** strategy is employed, where the model iteratively updates the latent state along a Markov chain based on the predicted noise to generate samples, defined as:

$$x_{t-1} = \frac{1}{\sqrt{\alpha_t}} \left(x_t - \frac{1-\alpha_t}{\sqrt{1-\bar{\alpha}_t}}\hat{\epsilon}_\theta(x_t, t)\right). \quad (5)$$

## 3. Methodology

We start with a brief overview of the proposed method, then provide a detailed explanation of its key stage.

## 3.1. Overview

To address the challenges of structural-semantic mismatch and dual-source noise interference within TAGs, we propose a novel uncertainty-modulated dual-path diffusion model, abbreviated as UDPD. The whole architecture is displayed in Figure 1, which consists of three key components. Specifically, we first employ a dual-perspective encoding strategy to separately extract semantic and structural embeddings. We then design a cooperative dual-path diffusion mechanism, where the structural and semantic branches operate in parallel and provide mutual conditioning. This interaction not only aligns their distributions but also purifies noise. Furthermore, we introduce a conditional mechanism modulated by node uncertainty. Functioning as an adaptive gate, it dynamically adjusts the strength of cross-branch interactions according to each node's uncertainty, thereby maximizing the denoising effectiveness on a per-node basis.

## 3.2. Dual-Perspective Node Encoding

Considering that TAGs mainly contain two types of essential information, namely node textual content and graph topology, we learn node embeddings from semantic and structural views, respectively.

**Semantic Embedding.** Given a node $v_i$, the associated raw textual description $t_i$ is typically lengthy and contains redundant information. To this end, we leverage a Large Language Model (LLM) (Wang et al., 2024) as a semantic filter to distill the most informative content. Specifically, we design a task-specific prompt that instructs the LLM to extract the Top-$K$ representative keywords from the raw text. Detailed prompt design is illustrated below:

> **System Prompt:** You are an expert in keyword extraction. Extract the most representative keywords from the raw text to capture its core semantics.
> **Instruction:** Extract the Top-$K$ keywords that best represent the node's category from the input text, separated by commas.
> **Example:**
> *Input:* "This paper proposes a new graph neural network architecture using attention mechanisms..."
> *Output:* Graph Neural Network, Attention Mechanism
> **Test Node:**
> *Input:* [The raw text $t_i$ of the target node $v_i$]
> *Output:* [Predict the keywords]

Mathematically, let $\mathcal{K}_i$ denote the extracted keyword set of node $v_i$, the extraction process can be represented as:

$$\mathcal{K}_i = \text{LLM}(t_i, \text{Prompt}), \quad (6)$$

In this way, the core semantic information is effectively distilled from the original text. Then, we employ a pretrained language encoder, e.g., Sentence-BERT (Reimers & Gurevych, 2019), to project the discrete keywords into a

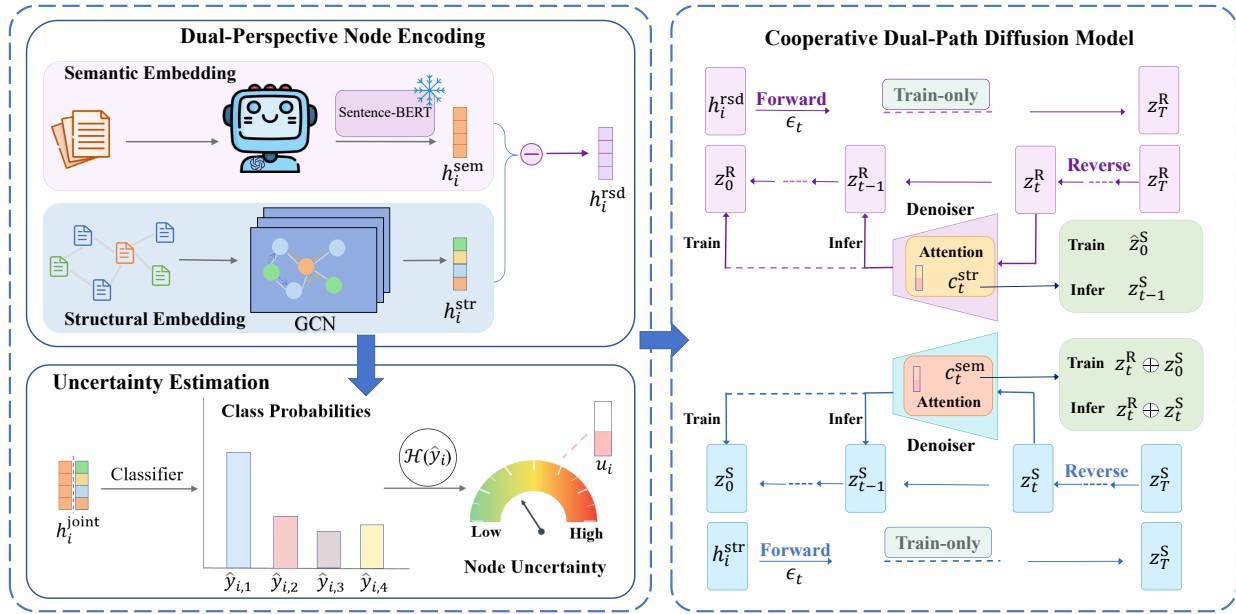

*Figure 1.* **The whole architecture of our proposed UDPD.**

continuous feature space, so as to obtain the node semantic embedding $e_i^{\text{sem}}$, defined as:

$$e_i^{\text{sem}} = \text{Encoder}(\mathcal{K}_i). \tag{7}$$

**Structural Embedding.** From a structural perspective, the goal is to capture the underlying topological patterns of nodes by aggregating information from their neighborhoods. Unlike semantic embedding, we initialize the structural embedding $h_i^{(0)}$ by encoding the raw text $t_i$, thereby preserving comprehensive semantic signals potentially relevant to structural connectivity, formalized as:

$$h_i^{(0)} = \text{Encoder}(t_i). \tag{8}$$

Then, the matrix form of initial structural embedding can be obtained as $H^{(0)}$. Here we adopt graph convolutional network (GCN) (Kipf & Welling, 2017) for information propagation, and the $(l+1)$-th layer embedding as follows:

$$H^{(l+1)} = \sigma\left(\tilde{D}^{-\frac{1}{2}}\tilde{A}\tilde{D}^{-\frac{1}{2}}H^{(l)}W^{(l)}\right), \tag{9}$$

where $\tilde{A} = A + I$ denotes the adjacency matrix with added self-loops, $\tilde{D} = \text{diag}(\tilde{d}_1, ..., \tilde{d}_n)$ is the diagonal degree matrix with $\tilde{d}_i = \sum_j \tilde{a}_{ij}$, $W^{(l)}$ represents the trainable weight matrix, and $\sigma(\cdot)$ is the non-linear activation function. In this way, we can acquire the structural embedding of target node $v_i$ from the final layer matrix $H^{(L)}$, represented as $e_i^{\text{str}}$.

**Semantic Residual Embedding.** To further distinguish a node's unique semantic nuances from shared neighborhood information, we introduce the concept of semantic residual. Structural embedding reflects shared neighborhood

characteristics but often overlook node-specific semantics, which are represented by semantic embedding. By referencing structural embedding, the semantic residual embedding highlights deviations from the neighborhood, emphasizing each node's unique semantic traits. Specifically, to ensure the embeddings have the same dimension, we first map the semantic embedding $e_i^{\text{sem}}$ and structural embedding $e_i^{\text{str}}$ into a common embedding space using two independent Multi-Layer Perceptrons (MLPs):

$$\begin{aligned} h_i^{\text{sem}} &= \text{MLP}_{\text{sem}}(e_i^{\text{sem}}), \\ h_i^{\text{str}} &= \text{MLP}_{\text{str}}(e_i^{\text{str}}). \end{aligned} \tag{10}$$

Then, we calculate the semantic residual embedding $h_i^{\text{rsd}}$ by directly subtracting the structural embedding from the semantic embedding as:

$$h_i^{\text{rsd}} = h_i^{\text{sem}} - h_i^{\text{str}}. \tag{11}$$

As a result, $h_i^{\text{rsd}}$ can effectively capture the personalized semantic information that makes the node different from its structural context.

### 3.3. Cooperative Dual-Path Diffusion Model

Here we propose a dual-path diffusion mechanism to simultaneously model the structural embedding $h_i^{\text{str}}$ and the semantic residual embedding $h_i^{\text{rsd}}$. Within this approach, both embeddings undergo a synchronized forward process, during which Gaussian noise is gradually injected into their initial states. The two paths then execute a conditional reverse process to progressively recover the clean embeddings.

**Forward Process.** For the structural branch, the structural

embedding $h_i^{\text{str}}$ serves as the initial state $z_0^{\text{S}}$, and the transition distribution for a single step is defined as:

$$q(z_t^{\text{S}} \mid z_{t-1}^{\text{S}}) = \mathcal{N}\left(z_t^{\text{S}}; \sqrt{1-\beta_t}\, z_{t-1}^{\text{S}}, \, \beta_t I\right). \quad (12)$$

Analogously, for the semantic residual branch, we set the semantic residual embedding $h_i^{\text{rsd}}$ as the initial state $z_0^{\text{R}}$, and apply the same noise injection process as:

$$q(z_t^{\text{R}} \mid z_{t-1}^{\text{R}}) = \mathcal{N}\left(z_t^{\text{R}}; \sqrt{1-\beta_t}\, z_{t-1}^{\text{R}}, \, \beta_t I\right). \quad (13)$$

**Reverse Process.** Given the noisy structural state $z_t^{\text{S}}$, we perform the reverse process to recover the clean signal as:

$$p_\theta(z_{t-1}^{\text{S}} \mid z_t^{\text{S}}) = \mathcal{N}\left(z_{t-1}^{\text{S}}; \mu_\theta(z_t^{\text{S}}, t), \Sigma_\theta(z_t^{\text{S}}, t)\right), \quad (14)$$

where $\mu_\theta$ and $\Sigma_\theta$ denote the mean and covariance predicted by the structural denoising network with parameter $\theta$.

Similarly, for the semantic residual branch, the reverse reconstruction process from $z_t^{\text{R}}$ is formalized as:

$$p_\psi(z_{t-1}^{\text{R}} \mid z_t^{\text{R}}) = \mathcal{N}\left(z_{t-1}^{\text{R}}; \mu_\psi(z_t^{\text{R}}, t), \Sigma_\psi(z_t^{\text{R}}, t)\right), \quad (15)$$

where $\mu_\psi$ and $\Sigma_\psi$ represent the mean and covariance estimated by the semantic residual denoising network with parameter $\psi$.

**Uncertainty-Modulated Condition Denoising.** Considering that effective denoising cannot rely solely on the independent refinement of each modality embedding, it also requires conditional guidance beyond the time step $t$ to adaptively regulate the interaction between structural and semantic embeddings (Li et al., 2025a). However, existing methods typically employ fixed interaction strategies, failing to account for the varying denoising requirements across nodes. To this end, inspired by (Zhang et al., 2025), we propose an uncertainty-modulated cross-attention strategy, which leverages node uncertainty as a gating signal to dynamically adjust the attention weights, thereby enabling "on-demand denoising" through flexible control of the interaction intensity between the two diffusion paths.

Specifically, given a node $v_i$, we first concatenate the aligned semantic embedding $h_i^{\text{sem}}$ and structural embedding $h_i^{\text{str}}$ to form a joint embedding $h_i^{\text{joint}}$, and predict the class distribution $\hat{y}_i$ via a classifier $f_\phi$. Based on the prediction, we can calculate the uncertainty $u_i$ as:

$$\hat{y}_i = f_\phi(h_i^{\text{joint}}), \quad (16)$$

$$u_i = -\frac{1}{\log C} \sum_{c=1}^{C} \hat{y}_{i,c} \log \hat{y}_{i,c}, \quad (17)$$

where $C$ denotes the total number of classes. Noting that the scalar $u_i \in [0,1]$ functions as an intensity factor in the

subsequent modules, which strengthens interaction to incorporate complementary information under high uncertainty, while suppressing it to mitigate noise interference when uncertainty is low.

**Structural Branch.** The structural branch employs a cross-attention strategy to incorporate information from the semantic domain. To maintain temporal consistency during the denoising process, we introduce a noise alignment strategy to construct the conditioning embedding $c_t^{\text{sem}} = z_0^{\text{S}} + z_t^{\text{R}}$, which combines the initial structural embedding with the current noisy semantic residual embedding.

Then, the structural denoising network $f_\theta$ adopts an uncertainty-guided attention mechanism to integrate semantic guidance. Formally, given current noisy structural state $z_t^{\text{S}}$, time step $t$, conditioning embedding $c_t^{\text{sem}}$, and node uncertainty $u_i$, the structural noise is predicted as:

$$\hat{\epsilon}_t^{\text{S}} = f_\theta(z_t^{\text{S}}, t, c_t^{\text{sem}}, u_i). \quad (18)$$

The cross-attention module (Vaswani et al., 2017) treats the structural embedding $z_t^{\text{S}}$ as the query, while the conditioning embedding $c_t^{\text{sem}}$ serves as both the key and value, with $u_i$ introduced as a modulation factor, represented as:

$$q_t^{\text{S}} = z_t^{\text{S}} W_q^{\text{S}}, \quad k_t^{\text{sem}} = c_t^{\text{sem}} W_k^{\text{S}}, \quad v_t^{\text{sem}} = c_t^{\text{sem}} W_v^{\text{S}},$$

$$o_t^S = (1 + u_i) \cdot \text{softmax}\left(\frac{q_t^{\text{S}}(k_t^{\text{sem}})^\top}{\sqrt{d_k}}\right) v_t^{\text{sem}}, \quad (19)$$

where $W_*^*$ denotes learnable linear projection.

Finally, based on the predicted noise $\hat{\epsilon}_t^{\text{S}}$, the clean structural state $\hat{z}_0^{\text{S}}$ is reconstructed via the inverse transformation:

$$\hat{z}_0^{\text{S}} = \frac{1}{\sqrt{\bar{\alpha}_t}}\left(z_t^{\text{S}} - \sqrt{1-\bar{\alpha}_t} \cdot \hat{\epsilon}_t^{\text{S}}\right). \quad (20)$$

**Semantic Residual Branch.** The semantic residual branch aims to recover high-frequency details that are not captured by structural smoothing, which are inherently more sensitive to noise. To this end, we introduce a stable structural reference strategy. Specifically, the clean structure embedding $\hat{z}_0^{\text{S}}$ reconstructed in the previous step is used as the conditioning embedding $c_t^{\text{str}}$, providing a reliable reference for semantic residual generation.

Then, the semantic denoising network $f_\psi$ uses the residual embedding $z_t^{\text{R}}$ as the Query to extract information from the conditioning embedding $c_t^{\text{str}}$, with $u_i$ incorporated as a modulation factor. The predicted noise is defined as:

$$\hat{\epsilon}_t^{\text{R}} = f_\psi(z_t^{\text{R}}, t, c_t^{\text{str}}, u_i), \quad (21)$$

where the cross-attention computation is given by:

$$q_t^{\text{R}} = z_t^{\text{R}} W_q^{\text{R}}, \quad k_t^{\text{str}} = c_t^{\text{str}} W_k^{\text{R}}, \quad v_t^{\text{str}} = c_t^{\text{str}} W_v^{\text{R}},$$

$$o_t^{\text{R}} = (1 + u_i) \cdot \text{softmax}\left(\frac{q_t^{\text{R}}(k_t^{\text{str}})^\top}{\sqrt{d_k}}\right) v_t^{\text{str}}. \quad (22)$$

Finally, the estimated initial semantic residual $\hat{z}_0^{\mathrm{R}}$ is derived via the inverse diffusion transformation as:

$$\hat{z}_0^{\mathrm{R}} = \frac{1}{\sqrt{\bar{\alpha}_t}} \left( z_t^{\mathrm{R}} - \sqrt{1 - \bar{\alpha}_t} \cdot \hat{\epsilon}_t^{\mathrm{R}} \right). \qquad (23)$$

In this way, the clean state estimates $\hat{z}_0^{\mathrm{S}}$ and $\hat{z}_0^{\mathrm{R}}$ are obtained, which are subsequently used to calculate the training objectives for parameter optimization.

**Training.** To train the dual-branch diffusion model, we adopt initial state reconstruction as the optimization objective, motivated by the nature of TAGs. Since nodes in TAG contain both high-dimensional semantic and structural patterns, directly reconstructing the initial state provides strong feature-level supervision. This objective encourages the model to accurately capture the underlying distributions of both semantic and structural information, ensuring that the generated embeddings preserve the original node identity and structural validity. The loss functions are defined as:

$$\mathcal{L}_{\mathrm{str}} = \mathbb{E}_{t, Z_0, \epsilon} \left[ \frac{1}{D} \left\| Z_0^{\mathrm{S}} - \hat{Z}_0^{\mathrm{S}} \right\|_2^2 \right], \qquad (24)$$

$$\mathcal{L}_{\mathrm{rsd}} = \mathbb{E}_{t, Z_0, \epsilon} \left[ \frac{1}{D} \left\| Z_0^{\mathrm{R}} - \hat{Z}_0^{\mathrm{R}} \right\|_2^2 \right], \qquad (25)$$

where $D$ is the feature dimension, and $\mathbb{E}$ denotes the expectation over time step $t$ and the Gaussian noise $\epsilon$.

### 3.4. Model Optimization

To support the downstream node classification task, we jointly optimize the diffusion objective together with a node classification loss. Mathematically, we first concatenate the generated semantic embedding $H^{\mathrm{sem}}$ and structural embedding $H^{\mathrm{str}}$ to form a joint embedding $H^{\mathrm{joint}} = [H^{\mathrm{sem}} \| H^{\mathrm{str}}]$. The predicted probability matrix $\hat{Y}$ is then calculated through a linear classifier:

$$\hat{Y} = \mathrm{softmax}(H^{\mathrm{joint}} W_{\mathrm{cls}}^{\top} + b_{\mathrm{cls}}). \qquad (26)$$

Then, we adopt the *label smoothing cross-entropy loss* (Guo et al., 2025) to reduce overfitting and handle label noise, thereby enhancing generalization by introducing a small amount of uncertainty to the target labels, which is denoted in the following as:

$$\mathcal{L}_{\mathrm{cls}} = -\frac{1}{|\mathcal{V}_{\mathrm{train}}|} \sum_{v \in \mathcal{V}_{\mathrm{train}}} \sum_{c=1}^{C} \left[ (1 - \gamma) y_{i,c} + \frac{\gamma}{C} \right] \log \hat{y}_{i,c}, \qquad (27)$$

where $\gamma \in [0, 1]$ is the smoothing factor, $y_{i,c}$ and $\hat{y}_{i,c}$ is the ground truth label and predicted probability, respectively.

Finally, the total loss $\mathcal{L}_{\mathrm{total}}$ can be obtained by combining the classification loss with the diffusion reconstruction losses, allowing the model to learn discriminative features while recovering data distributions:

$$\mathcal{L}_{\mathrm{total}} = \mathcal{L}_{\mathrm{cls}} + \lambda_1 \mathcal{L}_{\mathrm{str}} + \lambda_2 \mathcal{L}_{\mathrm{rsd}}, \qquad (28)$$

where $\lambda_1$ and $\lambda_2$ are hyperparameters balancing the weights.

### 3.5. Inference as Feature Purification

After training, we freeze the model parameters. Unlike standard generation from random noise, we propose a *feature purification* strategy. Specifically, we treat the imperfect raw features $h_i^{\mathrm{str}}$ and $h_i^{\mathrm{rsd}}$ as "noisy observations" of the ideal embeddings. They are mapped to the terminal diffusion state $T$, after which robust embeddings are recovered by filtering out noise through the reverse process.

Formally, we initialize the denoising chain as:

$$z_T^{\mathrm{S}} = h_i^{\mathrm{str}}, \quad z_T^{\mathrm{R}} = h_i^{\mathrm{rsd}}. \qquad (29)$$

For reverse steps $t = T, \dots, 1$, we use an alternating update strategy to exploit the interaction between the two branches.

**Structural Branch Update.** At each reverse diffusion step $t$, we first construct the semantic conditioning embedding $c_t^{\mathrm{sem}} = z_t^{\mathrm{S}} + z_t^{\mathrm{R}}$. The structural denoising network takes the current state $z_t^{\mathrm{S}}$, the conditioning embedding $c_t^{\mathrm{sem}}$, and the node uncertainty $u_i$ as inputs to predict the noise $\hat{\epsilon}_t^{\mathrm{S}}$. Using this predicted noise, a single-step update is performed to obtain the structural state at the previous time step $t - 1$, represented as:

$$z_{t-1}^{\mathrm{S}} = \frac{1}{\sqrt{\alpha_t}} \left( z_t^{\mathrm{S}} - \frac{1 - \alpha_t}{\sqrt{1 - \bar{\alpha}_t}} \hat{\epsilon}_t^{\mathrm{S}} \right). \qquad (30)$$

**Semantic Residual Branch Update.** The updated structural state $z_{t-1}^{\mathrm{S}}$ is subsequently adopted as the structure conditioning embedding $c_t^{\mathrm{str}}$, providing a cleaner structural guidance for the semantic residual branch. The residual denoising network takes this conditioning embedding as input to predict the noise $\hat{\epsilon}_t^{R}$, which is used to perform a single-step update of the residual state:

$$z_{t-1}^{\mathrm{R}} = \frac{1}{\sqrt{\alpha_t}} \left( z_t^{\mathrm{R}} - \frac{1 - \alpha_t}{\sqrt{1 - \bar{\alpha}_t}} \hat{\epsilon}_t^{\mathrm{R}} \right). \qquad (31)$$

The final outputs $z_0^{\mathrm{S}}$ and $z_0^{\mathrm{R}}$ at $t = 0$ serve as the purified embeddings for downstream tasks.

## 4. Experiments

To comprehensively evaluate the effectiveness of our proposed UDPD, we perform extensive experiments aimed at addressing the following research questions:

- **RQ1:** How does UDPD perform compared with state-of-the-art baselines?

- **RQ2:** How do different key components contribute to the overall performance of UDPD?
- **RQ3:** Can UDPD effectively resolve structural-semantic mismatch and yield discriminative node representations?

*Table 1.* Statistics of datasets.

| Datasets | #Classes | #Nodes | #Edges |
|---|---|---|---|
| Cora | 7 | 2,708 | 5,429 |
| Citeseer | 6 | 3,186 | 4,277 |
| PubMed | 3 | 19,717 | 44,338 |
| ArXiv-2023 | 40 | 46,198 | 78,543 |
| OGBN-Arxiv | 40 | 169,343 | 1,166,243 |

### 4.1. Experimental Setup

**Datasets.** We evaluate our UDPD on five widely used benchmark datasets, namely Cora (Sen et al., 2008), Citeseer (Giles et al., 1998), PubMed (Sen et al., 2008), ArXiv-2023 (He et al., 2024), and OGBN-ArXiv (Hu et al., 2020). These datasets are structured as citation graphs, where nodes represent academic papers and edges denote citation relationships between them. Each dataset is randomly partitioned into training, validation, and test subsets with a 6:2:2 ratio. Detailed dataset statistics are provided in Table 1.

**Baselines.** We compare UDPD against representative baselines from three categories, including **1) Traditional GNNs:** MLP, GCN (Kipf & Welling, 2017), GAT (Velickovic et al., 2018), GraphSAGE (Hamilton et al., 2017), and GCNII (Chen et al., 2020); **2) LM-based methods:** BERT (Devlin et al., 2019), GPT-4o (Hurst et al., 2024), and Gemini 2.5 Flash (Comanici et al., 2025); as well as **3) Recent works designed for TAGs:** G2P2 (Wen & Fang, 2023), GLEM (Zhao et al., 2023), SimTEG (Duan et al., 2023), TAPE (He et al., 2024), GraphBridge (Wang et al., 2024), CGT (Vakil & Amiri, 2024), LLM4NG (Yu et al., 2025), TANS (Wang et al., 2025d), and HiTuner (Fang et al., 2025).

**Implementation Details.** All methods are evaluated under the same data splits and reporting protocol for a fair comparison. For our proposed UDPD, we adopt a linear noise schedule ranging from $\beta_1 = 10^{-4}$ to $\beta_T = 0.02$, with the number of diffusion steps set to $T = 10$ by default ($T = 15$ for larger-scale datasets). For node textual content processing, we utilize Qwen2.5-7B-Instruct (Hui et al., 2024) to extract the top-$K$ keywords ($K = 10$) per node, and encode the resulting text using a frozen Sentence-BERT model. We set the loss balancing coefficients to $\lambda_1 = 0.5$ and $\lambda_2 = 0.5$ to balance the classification and diffusion objectives, and train the model for 500 epochs.

### 4.2. Performance Comparison

To evaluate the effectiveness of UDPD, we compare it against state-of-the-art baselines, reporting results averaged over five independent runs.

As shown in Table 2, we can find that UDPD consistently achieves the best performance across all datasets, verifying the effectiveness of our cooperative dual-path diffusion mechanism and uncertainty-modulated interaction strategy in resolving the structural-semantic mismatch and mitigating dual-source noise. Specifically, regarding traditional GNNs, methods like GAT rely heavily on deterministic message passing, which often struggles when facing significant structural-semantic mismatch. This limitation is particularly evident on the PubMed dataset, where node features are sparse and high-dimensional. While GAT only achieves an accuracy of 53.50%, UDPD reaches 94.93%, yielding a substantial improvement. This highlights that our dual-path approach effectively decouples the modalities, allowing the model to actively reconstruct valid signals from sparse inputs via iterative denoising, rather than passively accumulating noise through neighborhood aggregation. In addition, concerning LM-based methods, baselines that focus solely on semantic embeddings, such as BERT and GPT-4o, exhibit suboptimal performance due to their neglect of graph topology. For instance, on the highly connected OGBN-ArXiv dataset, GPT-4o only obtains 49.50%, lagging far behind UDPD (78.55%). This significant disparity confirms that relying exclusively on the semantic path is insufficient, underscoring the necessity of UDPD's design that jointly leverages structural and semantic pathways for effective node representation learning. Notably, UDPD outperforms recent TAG-specific methods, including advanced models like GraphBridge and TAPE, which rely on static feature fusion or sequential pipelines. On the Cora dataset, UDPD achieves 92.73%, surpassing GraphBridge (91.61%). This gain arises from our dual-path interactive idea, unlike prior approaches that unidirectionally inject semantic embedding into graph models, UDPD employs an uncertainty-modulated interaction in which structure calibrates semantic deviations, while semantic guide the refinement of noisy topology, yielding more robust cross-modal alignment.

### 4.3. Ablation Study

To validate UDPD's components, we evaluate five variants: 1) w/o Interaction (decoupled paths); 2/3) w/o Sem/Str-Diff (replacing semantic or structural diffusion with static inputs); 4) w/o Uncertainty (fixed interaction strength); and 5) w/o LLM-Emb (raw text encoding).

As shown in Table 3, our UDPD consistently outperforms all ablated variants. First, the dual-path mechanism is essential. Performance drops in w/o Sem/Str-Diff (e.g., 88.03% on Cora w/o Str-Diff and 74.88% on OGBN w/o Sem-Diff) reveal that the generative diffusion process is superior to relying on static observations for refining mismatched signals. Second, uncertainty modulation drives robustness. While

*Table 2.* **Experimental results of node classification.** We report the mean Accuracy (%) with a standard deviation of 5 runs with different random seeds. Highlighted are the top first, second, and third results.

| Model | Cora | Citeseer | PubMed | ArXiv-2023 | OGBN-ArXiv |
|---|---|---|---|---|---|
| MLP | $84.13 \pm 0.31$ | $69.99 \pm 0.31$ | $39.41 \pm 0.90$ | $70.20 \pm 0.33$ | $66.45 \pm 0.33$ |
| GCN | $84.97 \pm 0.34$ | $71.97 \pm 0.33$ | $50.11 \pm 2.04$ | $74.38 \pm 0.33$ | $70.76 \pm 0.16$ |
| GAT | $90.39 \pm 0.46$ | $72.74 \pm 0.42$ | $53.50 \pm 1.15$ | $70.46 \pm 0.20$ | $68.63 \pm 0.11$ |
| GraphSage | $85.56 \pm 1.54$ | $57.03 \pm 1.26$ | $82.35 \pm 0.60$ | $37.58 \pm 4.52$ | $32.22 \pm 5.49$ |
| GCNII | $88.51 \pm 0.52$ | $77.14 \pm 1.21$ | $89.54 \pm 0.35$ | $55.93 \pm 2.52$ | $58.36 \pm 2.73$ |
| BERT | $87.26 \pm 1.22$ | $72.88 \pm 0.35$ | $93.68 \pm 1.13$ | $58.97 \pm 2.39$ | $57.22 \pm 2.55$ |
| GPT-4o | $68.64 \pm 0.25$ | $68.75 \pm 0.31$ | $73.39 \pm 1.13$ | $64.86 \pm 1.22$ | $49.50 \pm 1.51$ |
| Gemini 2.5 Flash | $74.78 \pm 1.34$ | $66.46 \pm 1.71$ | $86.96 \pm 2.72$ | $66.58 \pm 2.65$ | $62.55 \pm 3.92$ |
| G2P2 | $83.26 \pm 2.39$ | $73.07 \pm 1.02$ | $81.29 \pm 0.11$ | $52.33 \pm 0.26$ | $63.89 \pm 0.01$ |
| SimTEG | $78.41 \pm 0.40$ | $66.19 \pm 0.29$ | $74.17 \pm 0.12$ | $63.31 \pm 0.14$ | $63.10 \pm 0.40$ |
| CGT | $89.26 \pm 0.45$ | $68.02 \pm 0.70$ | $78.72 \pm 0.55$ | $55.82 \pm 0.85$ | $57.40 \pm 0.77$ |
| GLEM | $87.11 \pm 1.14$ | $74.53 \pm 1.66$ | $85.24 \pm 2.70$ | $60.43 \pm 3.95$ | $63.35 \pm 0.21$ |
| TANS | $88.80 \pm 0.43$ | $77.30 \pm 0.91$ | $88.99 \pm 0.30$ | $67.78 \pm 0.38$ | $69.57 \pm 0.34$ |
| TAPE | $88.15 \pm 1.57$ | – | $94.67 \pm 0.48$ | $79.72 \pm 0.24$ | $77.02 \pm 0.23$ |
| GraphBridge | $91.61 \pm 1.86$ | $85.34 \pm 2.90$ | $82.97 \pm 1.69$ | $82.90 \pm 3.25$ | $76.03 \pm 0.12$ |
| Hituner | $88.14 \pm 0.96$ | $78.24 \pm 1.05$ | $94.64 \pm 0.29$ | – | – |
| LLM4NG | $87.55 \pm 1.17$ | $75.80 \pm 1.34$ | $88.61 \pm 0.22$ | $70.78 \pm 0.35$ | $73.67 \pm 0.26$ |
| **UDPD** | $\mathbf{92.73 \pm 0.58}$ | $\mathbf{85.89 \pm 1.12}$ | $\mathbf{94.93 \pm 0.50}$ | $\mathbf{83.34 \pm 0.92}$ | $\mathbf{78.55 \pm 1.31}$ |

w/o Interaction fails to correct biases (dropping to 82.72% on Citeseer), simply enabling fixed interaction (w/o Uncertainty) remains suboptimal compared to the full model (76.16% vs. 78.55% on OGBN). This proves that successful fusion requires adaptive regulation to filter noise based on node uncertainty. Third, the results for w/o LLM-Emb show that high-quality initialization amplifies diffusion efficacy, evidenced by a significant drop on PubMed (88.56% vs. 94.93%). Overall, improvements across five datasets validate that each module contributes uniquely to handling the complex challenges within TAGs.

*Table 3.* **Ablation study.** We report the mean classification accuracy (%) of UDPD and its variants over 5 runs across five datasets. The best results are highlighted in bold.

| Variants | Cora | Citeseer | PubMed | ArXiv | OGBN |
|---|---|---|---|---|---|
| w/o Interaction | 89.82 | 82.72 | 90.01 | 82.29 | 75.87 |
| w/o Sem-Diff | 89.89 | 84.40 | 89.09 | 81.03 | 74.88 |
| w/o Str-Diff | 88.03 | 84.11 | 89.24 | 80.94 | 75.98 |
| w/o Uncertainty | 88.50 | 84.08 | 89.46 | 82.71 | 76.16 |
| w/o LLM-Emb | 89.58 | 84.04 | 88.56 | 81.43 | 71.21 |
| **UDPD** | **92.73** | **85.89** | **94.93** | **83.34** | **78.55** |

### 4.4. Qualitative Analysis

To further gain an intuitive understanding of how UDPD refines node embeddings, we take the Cora dataset as an example, and visualize the denoising process in the feature space. Specifically, we extract the joint embeddings $H^{\mathrm{joint}}$ and project them into a 2D space using t-SNE (van der Maaten & Hinton, 2008).

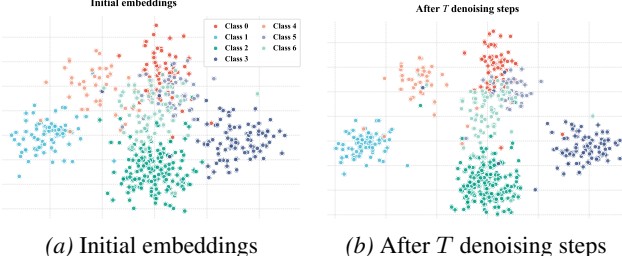

*(a)* Initial embeddings      *(b)* After $T$ denoising steps

*Figure 2.* Visualization of node embeddings on the Cora dataset before and after UDPD's denoising process.

As shown in Figure 2(a), before denoising, the joint embeddings $H^{\mathrm{joint}}$ exhibit a chaotic distribution, with nodes from different classes heavily intertwined and blurred decision boundaries. This reflects the challenge of structural-semantic mismatch, where raw signals contain significant noise that confuses the classifier. In contrast, Figure 2(b) highlights the effective purification capability of UDPD. After the iterative denoising process, the node embeddings evolve into distinct, compact clusters with high inter-class separability. This transition from chaos to order provides strong qualitative evidence that UDPD successfully filters out noise and rectifies misalignment, yielding discriminative embeddings readily separable by downstream classifiers.

## 5. Conclusion

In this paper, we present UDPD, a new uncertainty-modulated dual-path diffusion approach designed to address the persistent challenges of structural-semantic mismatch

and dual-source noise in TAGs. Specifically, our UDPD introduces a cooperative dual-path diffusion architecture. This allows for the independent refinement of semantic and structural embeddings while facilitating their progressive alignment through mutual guidance. Moreover, we incorporate an uncertainty-modulated condition denoising mechanism that dynamically calibrates interaction strength, ensuring robust cross-branch coupling. Extensive experiments on various benchmark datasets demonstrate that UDPD significantly outperforms state-of-the-art baselines, confirming its effectiveness and superiority.

## Acknowledgements

This work is supported by the National Natural Science Foundation of China (No. 62402337, No. 62422210, No. 62276187, No. 62272340, and No. 92370111) and the National Key Research and Development Program of China (No. 2023YFC3304503).

## Impact Statement

This paper presents work whose goal is to advance the field of machine learning. There are many potential societal consequences of our work, none of which we feel must be specifically highlighted here.

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

# A. Related Work

We concisely discuss some closely relevant researches, including representation learning on TAGs, diffusion models on graphs, as well as disentangled graph representation learning.

## A.1. Representation Learning on TAGs

Early approaches to learning on TAGs primarily adopt message passing mechanism that treat textual information merely as static initial features (Yang et al., 2015; Kipf & Welling, 2017). Representative models such as GCN, GAT, and GraphSAGE aggregate neighborhood information based on a fixed graph topology, using simple statistical features as node inputs (Kipf & Welling, 2017; Velickovic et al., 2018; Hamilton et al., 2017). Although effective for learning node representations, these methods often fail to capture the deep semantic dependencies inherent in complex textual attributes (Yang et al., 2021; Duan et al., 2023). To bridge this semantic gap, recent advancements have sought to integrate Pre-trained Language Models (PLMs) with GNNs (Devlin et al., 2019; Zhao et al., 2023). A standard method adopts a two-stage strategy, where PLMs are first fine-tuned to generate high-quality node features, which are then fed into GNNs (Devlin et al., 2019; Duan et al., 2023). For instance, SimTEG leverages parameter-efficient tuning to adapt LMs to graph-structured data (Duan et al., 2023). Another line of research explores joint training strategies to enable deeper modality interaction (Zhao et al., 2023; He et al., 2024). LLM4NG adapts Large Language Models directly for graph inference tasks, while GLEM employs a variational Expectation-Maximization (EM) framework to alternate between optimizing the LM and GNN modules (Yu et al., 2025; Zhao et al., 2023). Despite their performance gains, these approaches typically enforce a rigid fusion of modalities (Wang et al., 2024; Zhou et al., 2025b). They operate under the assumption that structural connections and semantic content are consistent, thereby ignoring the common structural-semantic mismatch in real-world graphs (Zhu et al., 2020; Wang et al., 2025a; Linkerhägner et al., 2025). Moreover, most existing methods lack dynamic mechanisms to filter dual-source noise (e.g., irrelevant edges or ambiguous text), often leading to interference during the fusion process (Zhang & Zitnik, 2020; Linkerhägner et al., 2025; Zhou et al., 2025a).

## A.2. Diffusion Models on Graphs

Generative diffusion models have revolutionized data synthesis in computer vision and natural language processing by learning to progressively reverse a noise corruption process (Ho et al., 2020). While there have been initial attempts to extend these methods to the graph domain, current research predominantly focuses on graph generation tasks, such as molecular design or topology synthesis (Jo et al., 2022; Vignac et al., 2023). These methods are typically tailored for generating discrete structures or simple attributes, aiming to capture geometric constraints rather than semantic representations (Austin et al., 2021; Vignac et al., 2023). In contrast, the potential of diffusion models for representation learning on TAGs remains a largely uncharted frontier (Jo et al., 2022; Vignac et al., 2023; Zhao et al., 2023). Unlike molecular graphs, TAGs typically contain rich semantic information within nodes, which introduces a complex interplay between textual content and graph structure (Yang et al., 2021; Zhao et al., 2023; Wang et al., 2024; Duan et al., 2023). Existing graph diffusion approaches struggle to model such high-dimensional semantic distributions and lack explicit mechanisms for aligning structural and semantic information (Jo et al., 2022; Vignac et al., 2023; Zhu et al., 2020; Wang et al., 2025a). As a result, there is a distinct scarcity of approaches leverage the generative denoising paradigm to address the challenging structural-semantic misalignment inherent in TAGs (Ho et al., 2020; Jo et al., 2022; Vignac et al., 2023; Zhu et al., 2020; Wang et al., 2025a).

## A.3. Disentangled Graph Representation Learning

Disentangled representation learning focuses on separating the distinct, informative factors of variations hidden in the data, a concept that has shown great promise in enhancing model interpretability and robustness (Higgins et al., 2017). In the graph domain, pioneering researches involving *decoupled* designs and denoising mechanisms have attempted to extend this paradigm, e.g., decoupling the dependency between feature transformation and propagation, or separating clean structural signals from noise (Wu et al., 2019). These approaches typically aim to decouple prediction/feature transformation from propagation, or to separate reliable connections from interference via structure-aware defense/denoising (Rong et al., 2020). However, current studies predominantly concentrate on *intra-modal* factors, that is, decomposing the structural signal itself, while largely overlooking the *inter-modal* entanglement between graph topology and node textual semantics (Zhang et al., 2025). In TAGs, the distribution of node textual semantics often diverges from the graph topology, creating a structural-semantic misalignment (Li et al., 2025b). Existing methods rarely explicitly disentangle these two modalities during the learning process, leading to representations where semantic and structural signals are inextricably mixed (Su

et al., 2025). This lack of clear modality disentanglement limits the model's ability to selectively filter noise specific to each source, underscoring the need for a architecture that can independently refine and subsequently align these distinct data views (Zhao et al., 2023).

## B. Analysis and Statistics of Structural-Semantic Mismatch

In this section, we provide deeper empirical insights and rigorous quantitative evidence to justify our core motivation regarding the prevalence of structural-semantic mismatch in TAGs. While standard graph neural networks inherently assume topological-semantic homophily, we demonstrate both statistically and taxonomically that this underlying assumption is frequently violated across standard datasets.

### B.1. Quantitative Distributional Divergence

To quantitatively assess the discrepancy between graph topology and textual semantics, we compute two key metrics across five datasets: (1) **Semantic Similarity**, measured via the cosine similarity of neighbor node embeddings encoded by Sentence-BERT, and (2) **Jensen-Shannon Divergence (JSD)**, which quantifies the statistical divergence between the neighborhood topological distributions and the text-derived semantic clusters.

As illustrated in Table 4, neighboring nodes across all datasets exhibit relatively low semantic similarity, with PubMed exhibiting the lowest score ($0.356$). Concurrently, the JSD values consistently exceed $0.4$, reaching a peak of $0.727$ on PubMed. These statistics provide robust empirical evidence that interconnected nodes in TAGs do not necessarily share well-aligned semantic spaces, thereby substantiating the necessity of our dual-path cooperative diffusion mechanism.

*Table 4.* Quantitative statistics of structure-semantic mismatch on actual datasets.

| Metrics | Cora | Citeseer | PubMed | ArXiv-2023 | OGBN-ArXiv |
|---|---|---|---|---|---|
| Semantic Similarity | 0.480 | 0.563 | 0.356 | 0.546 | 0.454 |
| Jensen-Shannon Divergence (JSD) | 0.511 | 0.509 | 0.727 | 0.557 | 0.419 |

### B.2. A Taxonomical Typology of Mismatched Edges

Through a granular review of the connected node pairs, we categorize the observed structural-semantic mismatch into four primary types, heavily drawing upon our empirical observations and verified statistics:

- **Different Topics:** In broad datasets such as OGBN-ArXiv and ArXiv-2023, cross-field connections frequently occur. As noted in our previous investigation, a computer vision paper may cite an NLP paper to reuse a model or methodology. Although a structural edge exists, their textual semantics may differ substantially. Quantitatively, this type accounts for $26.86\%$ of connected edges in OGBN-ArXiv and $24.40\%$ in ArXiv-2023.

- **Granular Focus: The Same Topic but Different Aspects:** In datasets like Cora and Citeseer, connected nodes often share a high-level research area but differ significantly in their specific focus. For instance, two papers in Cora may both belong to "Reinforcement Learning", yet one focuses on theoretical analysis while the other studies robotic control applications. This type accounts for $23.59\%$ of connected edges in Cora and $19.57\%$ in Citeseer.

- **Semantic Mismatch in Specialized Domains:** In domain-specific datasets like PubMed, papers often rely on tools, evidence, or findings from different subtopics. As a result, structurally connected papers may still exhibit substantial semantic differences. In PubMed, $35.17\%$ of connected edges fall into the different topics category, while $39.90\%$ correspond to the same topic but different aspects category, indicating that semantic mismatch is also prevalent in specialized domains.

- **Direct Contradiction or Inconsistencies:** Across all datasets, a structural edge does not necessarily imply agreement between two papers. For instance, a recent work may cite a previous method primarily to highlight its limitations. If a graph model directly aggregates features from such semantically inconsistent nodes, it may introduce harmful noise. Our analysis using Natural Language Inference (NLI) to detect textual contradictions further supports this observation, identifying potentially contradictory or inconsistent edges in $26.17\%$ of connected pairs in Cora, $23.98\%$ in Citeseer, $39.10\%$ in PubMed, $33.60\%$ in OGBN-ArXiv, and $33.92\%$ in ArXiv-2023.

These four types highlight the complex interplay between graph structure and textual semantics in real-world graphs. Our proposed method is designed to effectively address such structural-semantic mismatch.

## C. Theoretical Background on Diffusion Models

Here we provide the theoretical formulation of Denoising Diffusion Probabilistic Models (DDPMs) (Ho et al., 2020; Zhou et al., 2024). Formally, we use $x$ to denote the random variable from the data distribution. In the context of our proposed UDPD, $x$ corresponds to the latent embeddings $z = \{z^{\text{S}}, z^{\text{R}}\}$ derived from the structural and semantic residual encoders, respectively.

### C.1. Forward Diffusion Process

Given a data point $x_0$ sampled from the ground truth data distribution $q(x_0)$, the forward diffusion process is defined as a fixed Markov chain that gradually adds Gaussian noise to the data according to a pre-defined variance schedule $\{\beta_1, \ldots, \beta_T\}$. Formally, the joint distribution of the forward process is defined as:

$$q(x_{1:T}|x_0) := \prod_{t=1}^{T} q(x_t|x_{t-1}), \tag{32}$$

where each transition step $q(x_t|x_{t-1})$ is a Gaussian distribution:

$$q(x_t|x_{t-1}) := \mathcal{N}(x_t; \sqrt{1-\beta_t}x_{t-1}, \beta_t I). \tag{33}$$

A key property of this parameterization is that it allows sampling $x_t$ at an arbitrary timestep $t$ directly from $x_0$ in closed form. Mathematically, let $\alpha_t := 1 - \beta_t$ and $\bar{\alpha}_t := \prod_{s=1}^{t} \alpha_s$, the marginal distribution $q(x_t|x_0)$ can be expressed as:

$$q(x_t|x_0) = \mathcal{N}(x_t; \sqrt{\bar{\alpha}_t}x_0, (1-\bar{\alpha}_t)I). \tag{34}$$

Using the reparameterization trick, $x_t$ can be sampled as:

$$x_t = \sqrt{\bar{\alpha}_t}x_0 + \sqrt{1-\bar{\alpha}_t}\epsilon, \quad \text{where } \epsilon \sim \mathcal{N}(0, I). \tag{35}$$

As $t \to T$, $\bar{\alpha}_T$ approaches 0, and the distribution $q(x_T|x_0)$ converges to a standard isotropic Gaussian $\mathcal{N}(0, I)$.

### C.2. Reverse Generative Process

The generative process aims to reverse the forward diffusion chain to recover $x_0$ from pure Gaussian noise $x_T \sim \mathcal{N}(\mathbf{0}, \mathbf{I})$. We approximate the intractable posterior $q(x_{t-1}|x_t)$ using a parameterized model $p_\theta$:

$$p_\theta(x_{0:T}) := p(x_T) \prod_{t=1}^{T} p_\theta(x_{t-1}|x_t), \tag{36}$$

where the transition is modeled as a Gaussian $p_\theta(x_{t-1}|x_t) := \mathcal{N}(x_{t-1}; \mu_\theta(x_t, t), \Sigma_\theta(x_t, t))$.

### C.3. Optimization Objective: From Noise Prediction to Feature Reconstruction

The training of diffusion models is based on maximizing the Variational Lower Bound (VLB) (Huang et al., 2021; Kingma & Welling, 2014) on the log-likelihood of the data, denoted as:

$$\mathbb{E}_{q(x_0)}[\log p_\theta(x_0)] \geq \mathbb{E}_{q(x_{0:T})}\left[\log p(x_T) + \sum_{t=1}^{T} \log \frac{p_\theta(x_{t-1}|x_t)}{q(x_t|x_{t-1})}\right]. \tag{37}$$

To optimize this objective efficiently, Ho et al. (2020) simplified the parameterization by training the model to predict the added noise $\epsilon$. This leads to the standard "simple" loss function (Nichol & Dhariwal, 2021; Rombach et al., 2022):

$$\mathcal{L}_{\text{noise}} = \mathbb{E}_{x_0, \epsilon, t}\left[\|\epsilon - \epsilon_\theta(x_t, t)\|^2\right]. \tag{38}$$

**Alternative Parameterization (Feature Reconstruction).** Although predicting noise is standard for image generation, it is mathematically equivalent to predicting the original signal $x_0$. In our model, since we aim to involve explicitly aligning structural and semantic embeddings, we adopt the $x_0$-prediction parameterization (i.e., predicting the clean latent $z_0$). The relationship between the predicted noise $\epsilon_\theta$ and the predicted data $\hat{x}_0$ is given by Eq. (4): $\hat{x}_0 = \frac{1}{\sqrt{\bar{\alpha}_t}}(x_t - \sqrt{1 - \bar{\alpha}_t}\epsilon_\theta)$. Consequently, the training objective is formulated as a *reconstruction loss* in the latent space:

$$\mathcal{L}_{\text{recon}} = \mathbb{E}_{x_0, \epsilon, t}\left[\|x_0 - \hat{x}_\theta(x_t, t)\|^2\right]. \tag{39}$$

This encourages the model to directly recover the clean topological and semantic signals from their noisy counterparts, facilitating more effective cross-modal alignment.

## D. Theoretical Justification of UDPD

### D.1. Effectiveness of Applying Continuous Diffusion to TAGs

A primary challenge in applying diffusion models to TAGs stems from the discrete nature of graph topology and the high-dimensional nature of raw features. Standard DDPMs are formulated under the assumption that data reside in a continuous Euclidean space and and evolve according to Gaussian transition kernels, which makes them theoretically ill-suited for discrete adjacency matrices $A \in \{0, 1\}^{N \times N}$ or discrete text tokens.

For our UDPD, we address this challenge by operating within a continuous latent space. Given the text-attributed graph $\mathcal{G}$, instead of performing diffusion directly on $\mathcal{G}$, we first obtain initial semantic and structural embeddings, denoted as $e_i^{\text{sem}}$ and $e_i^{\text{str}}$, respectively. These embeddings are then projected into a unified continuous Euclidean latent space $H \subseteq \mathbb{R}^d$ via MLP-based projections. This transformation ensures that the resulting representations satisfy the continuity assumption required for Gaussian diffusion processes.

**Proposition D.1.** *(Manifold Mapping). Assuming that the graph generation process follows a latent variable model, where the observed discrete graph $\mathcal{G}$ is generated from continuous latent variables $h = \{h_i^{\text{str}}, h_i^{\text{sem}}\} \in H$, derived from $e_i^{\text{str}}$ and $e_i^{\text{sem}}$. Under this assumption, optimizing the diffusion model in the latent space $H$ naturally permits the use of standard Gaussian transition kernels on continuous representations. This provides a theoretical justification for employing continuous diffusion formulations on discrete graph data, thereby circumventing the need for discrete noise definitions.*

### D.2. Dual-Path Evidence Lower Bound

Unlike single-modal diffusion methods, our proposed UDPD aims to model the joint distribution of structural and semantic residual embeddings $p(z^{\text{S}}, z^{\text{R}})$. Although the two diffusion processes are independently parameterized, their conditional interactions enable effective optimization of a joint variational lower bound (VLB) (Sohl-Dickstein et al., 2015; Kingma et al., 2021). Mathematically, the log-likelihood of the joint data can be bounded as:

$$\log p(z_0^{\text{S}}, z_0^{\text{R}}) \geq \mathbb{E}_q\left[\sum_{t=1}^{T} \log \frac{p_\theta(z_{t-1}^{\text{S}}, z_{t-1}^{\text{R}}|z_t^{\text{S}}, z_t^{\text{R}})}{q(z_t^{\text{S}}, z_t^{\text{R}}|z_{t-1}^{\text{S}}, z_{t-1}^{\text{R}})}\right]. \tag{40}$$

Then, we factorize the reverse process using an uncertainty-modulated interaction mechanism. Assuming conditional independence given the interaction term, the joint reverse transition is approximated as:

$$p_\theta(z_{t-1}^{\text{S}}, z_{t-1}^{\text{R}}|z_t^{\text{S}}, z_t^{\text{R}}) \approx p_\theta(z_{t-1}^{\text{S}}|z_t^{\text{S}}, o_t^{\text{S}}) \cdot p_\theta(z_{t-1}^{\text{R}}|z_t^{\text{R}}, o_t^{\text{R}}), \tag{41}$$

where $o_t^{\text{S}}$ and $o_t^{\text{R}}$ denote the uncertainty-modulated cross-attention outputs defined in Eq. (19) and Eq. (22). These terms act as the coupling statistics that inject the semantic embedding into the structural embedding (and vice versa) based on the estimated node uncertainty (Rombach et al., 2022; Zhai et al., 2024).

This factorization constitutes the theoretical foundation of our model. It demonstrates that exchanging uncertainty-modulated contextual information via cross-attention effectively enables the optimization of the joint distribution over the structural and semantic modalities (Ruan et al., 2023). Consequently, the proposed dual-path strategy is not merely an architectural choice, but a principled variational approximation to joint log-likelihood maximization, ensuring that the structural and semantic embeddings are learned in a mutually consistent manner.

# E. Theoretical Analysis of Uncertainty Quantification

To effectively guide the dual-path diffusion process, it is crucial to identify nodes graph structure conflicts with node textual content. While uncertainty may arise from various sources, recent studies suggest that comprehensive uncertainty quantification on graphs should account for both *Prediction Ambiguity* (Label Entropy) and *Topological Dissonance* (Label Disharmonicity). To this end, we provide a theoretical justification showing that the simple entropy metric can serve as an efficient and unified proxy for these complex uncertainty factors.

## E.1. Entropy as a Surrogate for Topological Dissonance

Ideally, structurally uncertain nodes should be characterized by *Label Disharmonicity*, which measures the discrepancy between a node's prediction and the consensus of its local neighborhood (Zhang et al., 2025). Formally, given a node $v_i$, we define its structural dissonance score $\mathcal{D}_{\text{struct}}(v_i)$ as:

$$\mathcal{D}_{\text{struct}}(v_i) = \left\| \hat{y}_i - \frac{1}{|N(v_i)|} \sum_{v_j \in N(v_i)} \hat{y}_j \right\|_2^2, \tag{42}$$

where $\hat{y}_i$ represents the predicted class probability distribution for node $v_i$. Although $\mathcal{D}_{\text{struct}}(v_i)$ provides a principled notion of structural conflict (e.g., identifying heterophilous connections (Pei et al., 2020)), computing it explicitly requires repeated neighborhood propagation and aggregation, which is computationally expensive during the diffusion training loop.

We therefore employ *predictive entropy* as an efficient surrogate within the Message-Passing Neural Network (MPNN) framework (Gilmer et al., 2017). The rationale is grounded in the aggregation mechanism of GNNs, that is, nodes with high topological disharmonicity tend to aggregate mutually inconsistent evidence from their neighbors (e.g., a node of class A surrounded by neighbors of class B and C). This conflicting message passing produces a "flatter" (less confident) posterior distribution $\hat{y}_i$, naturally resulting in higher predictive entropy without explicit neighborhood computation (Gal et al., 2017; Gal & Ghahramani, 2016).

## E.2. Unified Uncertainty Lower Bound

We further justify this surrogate choice by analyzing the information-theoretic properties of the aggregation process. Specifically, for a broad class of attention-based GNNs (Velickovic et al., 2018), the predictive entropy of a node $v_i$ admits a lower bound that is jointly determined by the entropy of its neighbors and the structural dissonance (Cover & Thomas, 2006), denoted as:

$$H(\hat{y}_i) \geq \sum_{v_j \in N(v_i)} \eta_{ij} H(\hat{y}_j) + \lambda \cdot \mathcal{D}_{\text{struct}}(v_i), \tag{43}$$

where $\eta_{ij}$ denotes the normalized attention weights between nodes $v_i$ and $v_j$, and $\lambda > 0$ is a scaling factor determined by the convexity of aggregation function (Choromanska & Jain, 2019).

This inequality has a profound implication: an increase in structural disharmony $\mathcal{D}_{\text{struct}}(v_i)$ necessarily raises the lower bound of the node's predictive entropy $H(\hat{y}_i)$. Consequently, by adopting entropy as our uncertainty indicator $u_i$, we simultaneously capture both intrinsic ambiguity (the first term, derived from noisy features) and structural-semantic mismatch (the second term, derived from topological dissonance) (Zhang et al., 2025), achieving a unified uncertainty quantification with $O(1)$ computational complexity.

## E.3. Implementation Strategy

Based on the derivation above, we define the robust uncertainty score $u_i$ using the normalized entropy metric:

$$u_i = -\frac{1}{\log C} \sum_{c=1}^{C} \hat{y}_{i,c} \log \hat{y}_{i,c}, \tag{44}$$

where $\hat{y}_{i,c}$ denotes the predicted probability of node $v_i$ belonging to class $c$. The normalization ensures $u_i \in [0, 1]$, where $u_i = 0$ indicates high confidence, and $u_i = 1$ implies a uniform (maximally uncertain) distribution.

To balance adaptivity with training stability, we adopt an epoch-wise update strategy, i.e., the node uncertainty $u_i$ is re-evaluated only at the beginning of each training epoch to reflect the evolving quality of node embeddings, and kept fixed

throughout the multi-step diffusion trajectory within that epoch. This ensures a consistent uncertainty signal for the step-wise denoising process (as in Eq. 19 and 22), while allowing the estimates to progressively refine as the model converges.

# F. Additional Experimental Results

### F.1. Sensitivity Analysis of Keyword Number $K$

In the semantic initialization phase, the hyperparameter $K$ determines the number of representative keywords extracted from the raw textual contents by the LLM. To evaluate the impact of keyword granularity on model performance, we vary $K \in \{5, 10, 20\}$ and visualize the results in Figure 3.

As illustrated, UDPD exhibits a consistent performance trend across all five datasets. When $K = 5$, the performance is relatively low, indicating that extracting too few keywords leads to semantic scarcity, where critical information required for node classification is lost. In contrast, increasing $K$ to 20 results in a performance degradation compared to $K = 10$. This suggests that an excessive number of keywords leads to information redundancy, which potentially diminishes the distinctiveness of the node embeddings by including less discriminative terms. Overall, the experimental results confirm that setting $K = 10$ achieves an optimal balance, effectively capturing core semantics while avoiding redundancy.

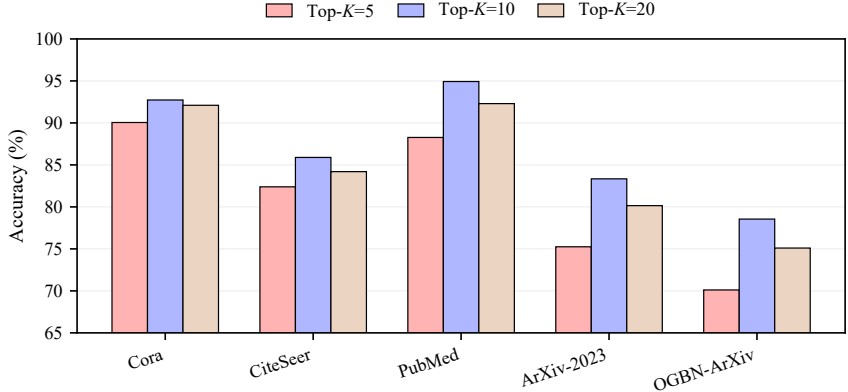

*Figure 3.* Performance comparison with different numbers of extracted keywords $K$.

### F.2. Dynamics of Node Uncertainty during Training

To validate the dynamic nature of our uncertainty-modulated mechanism, we track the average node uncertainty across five datasets over the course of training, from epoch 0 to epoch 500. The results, illustrated in Figure 4, reveal a consistent declining trend in node uncertainty, effectively demonstrating the adaptive nature of the learning process.

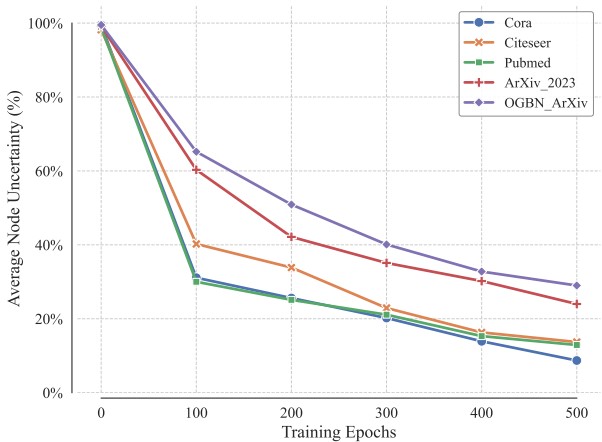

*Figure 4.* Dynamics of average node uncertainty from epoch 0 to epoch 500.

As observed, node uncertainty is initially high ($\approx 99\%$) at the early stage of training (Epoch 0). Within our UDPD architecture, this high uncertainty serves as an active control signal that intensifies the interaction between the structural branch and the semantic residual branch. By strengthening cross-modal coupling, the model is encouraged to exploit complementary information from both modalities, thereby correcting severe structural–semantic mismatch present in the initial representations.

As training progresses (epoch 100-300), node uncertainty decreases rapidly, indicating that the model is successfully resolving the topological dissonance. Crucially, during the final convergence phase (epoch 400-500), node uncertainty stabilizes at a consistently low level (e.g., 8.70% on Cora). This reduction is not merely a sign of convergence, but functionally triggers the gating mechanism to suppress further cross-modal interaction. Such an adaptive suppression mechanism is essential, as it prevents the model from over-assimilating irrelevant neighborhood noise once robust representations have been formed, effectively transitioning the model's focus from "global information exploration" to "local feature refinement".

