# OpenReview forum: "Bridging Structure and Semantics: Uncertainty-Modulated Dual-Path Diffusion for Robust Text-Attributed Graph Learning"
_ICML.cc/2026/Conference — ICML 2026 regular_

### Official Review · Reviewer_1d4Y · 2026-03-07

**Soundness:** 3
**Presentation:** 3
**Significance:** 3
**Originality:** 3
**Overall Recommendation:** 5
**Confidence:** 4

**Summary:**

This paper introduces a new approach called UDPD for robust representation learning on text-attributed graphs. The authors clearly point out the limits of current methods when dealing with mismatched graph structures and node semantics. By operating structural and semantic diffusion branches in parallel and introducing an uncertainty-based conditional cross-attention mechanism, UDPD achieves effective alignment and denoising of the two modalities.

**Compliance With Llm Reviewing Policy:**

Affirmed.

**Final Justification:**

The authors’ response has addressed my concerns, and I will maintain my score.

**Key Questions For Authors:**

1. Could the authors provide a quantitative comparison of the computational overhead (i.e., training time, inference latency, and memory consumption) against top-performing baselines like GraphBridge or GLEM?
2. How much does the model's performance change if you use a different LLM (like replacing Qwen2.5-7B with a smaller model) or change the prompt for extracting keywords?
3. If the text or graph inputs had a lot more noise (like adversarial or highly heterophilous data), would the uncertainty modulation and denoising still work well?

**Limitations:**

Yes

**Strengths And Weaknesses:**

Strengths:
1. This paper proposes a new dual-path fusion mechanism guided by uncertainty. Using continuous diffusion on discrete text-attributed graphs is a great way to fix the mismatch between structure and semantics, opening up a promising research direction.
2. The paper is well-written, clearly structured, and easy to follow. The model design is presented in a straightforward and understandable manner.
3. The experiments are convincing. The model consistently beats SOTA baselines across five benchmark datasets. Furthermore, the detailed ablation studies clearly show that both the dual-path design and the uncertainty modules are necessary.

Weaknesses:
1. The dual-path diffusion and cross-attention mechanisms naturally add some extra computing cost. It would be great to see a detailed comparison of runtime, training time, and memory usage against the baselines.
2. The paper highlights the model's robustness to dual-source noise, but the tests mostly rely on citation graphs. To really prove its general robustness, it would be necessary to test it on datasets with strong heterophily, or by purposely adding synthetic and adversarial noise.
3. The model relies heavily on external LLMs (like Qwen2.5-7B) to extract keywords, but this isn't explored in depth. Analyzing the model's sensitivity to different LLMs or variations in prompt quality would significantly strengthen the claims regarding its robustness.
4. Even though using predictive entropy makes sense in theory, comparing it with practical alternatives like dropout-based uncertainty could give us deeper insights.

---

> ### Author Rebuttal · Authors · 2026-03-31
>
> **W1&Q1**: Following your suggestion, we have added a quantitative comparison of training and inference time. As shown in Table 1, on the largest OGBN-ArXiv dataset, the training time of our UDPD in roughly 25 minutes. In comparison, baselines such as GraphBridge and GLEM typically require 1-2 days to complete training, highlighting the computational efficiency of UDPD. Regarding memory usage, our model reaches a peak GPU memory usage of 20.9 GB on OGBN-ArXiv, which is comparable to that of the baselines. This demonstrates that our method can achieve high performance without incurring additional memory overhead.
>
> Table 1. Training and Inference Time of Our Model
>
> | Datasets    | Training Time (s) | Inference Time (s) |
> | :--------- | :---------------: | :----------------: |
> | Cora       | 69.43±5.44        | 29.14±1.81         |
> | Citeseer   | 36.30±3.36        | 17.10±1.15         |
> | PubMed     | 473.93±6.60       | 177.98±2.29        |
> | ArXiv-2023 | 415.47±6.55       | 158.49±3.25        |
> | OGBN-ArXiv | 1517.23±2.13      | 794.60±2.10        |
>
>
> **W2&Q3**: Thanks! We have added experiments with synthetic structural noise to evaluate UDPD’s robustness. As shown in Table 2, as the level of injected structural noise increases, our UDPD consistently maintains higher performance than GraphBridge. Even under 0.3 structural noise, on Cora and Citeseer datasets, UDPD’s performance drops by 4.17% and 2.81%, respectively, whereas GraphBridge suffers larger declines of 8.56% and 7.11%, highlighting the robustness of our method.
>
> Table 2. Performance Comparison under Injected Structural Noise
>
> | Datasets  | Methods      | Clean          | Noise=0.1      | Noise=0.2      | Noise=0.3      |
> | :------- | :---------- | :------------: | :------------: | :------------: | :------------: |
> | Cora     | UDPD (Ours) | **92.73±0.58** | **91.16±3.19** | **89.94±0.21** | **88.56±0.80** |
> | Cora     | GraphBridge | 91.61±1.86     | 87.27±0.76     | 86.51±1.35     | 83.05±2.27     |
> | Citeseer | UDPD (Ours) | **85.89±1.12** | **84.68±0.89** | **83.95±0.90** | **83.08±1.46** |
> | Citeseer | GraphBridge | 85.34±2.90     | 82.85±1.93     | 79.72±2.05     | 78.23±2.77     |
>
>
> **W3&Q2**: Thanks! We have added experiments to evaluate how the model’s performance changes when replacing Qwen2.5-7B with a smaller model (Qwen2.5-1.5B) or a different LLM (LLaMA-3-8B). As shown in Table 3, replacing Qwen2.5-7B with a smaller model (Qwen2.5-1.5B) results in only a minor performance drop, while using LLaMA-3-8B achieves nearly identical results. This demonstrates that the performance improvements are primarily driven by our dual-path diffusion mechanism rather than by a specific large LLM.
>
> Table 3. Performance Comparison with Different LLMs
>
> | Variants              | Cora           | Citeseer       | PubMed         | ArXiv-2023     | OGBN-ArXiv     |
> | :------------------- | :------------: | :------------: | :------------: | :------------: | :------------: |
> | Qwen2.5-7B (Default) | **92.73±0.58** | **85.89±1.12** | **94.93±0.50** | 83.34±0.92     | **78.55±1.31** |
> | Qwen2.5-1.5B         | 91.25±0.22     | 84.21±0.36     | 93.29±0.17     | 81.87±1.26     | 76.86±2.15     |
> | LLaMA-3-8B           | 92.49±0.18     | 85.65±0.95     | 93.80±1.13     | **83.39±0.33** | 78.10±0.97     |
>
>
>
> **W4**: Following your suggestion, we have added experiments comparing predictive entropy with the practical alternative, MC Dropout. As shown in Table 4, replacing predictive entropy with MC Dropout results in a clear performance drop across all five datasets, demonstrating that predictive entropy is better suited for our UDPD.
>
> Table 4. Performance Comparison: Predictive Entropy vs. MC Dropout
>
> | Methods             | Cora           | Citeseer       | PubMed         | ArXiv-2023     | OGBN-ArXiv     |
> | :----------------- | :------------: | :------------: | :------------: | :------------: | :------------: |
> | MC Dropout         | 88.32±2.55     | 82.67±0.31     | 92.50±1.37     | 78.29±0.42     | 73.01±2.21     |
> | Predictive Entropy | **92.73±0.58** | **85.89±1.12** | **94.93±0.50** | **83.34±0.92** | **78.55±1.31** |

---

> > ### Author Rebuttal · Reviewer_1d4Y · 2026-04-01
> >
> > The authors’ response has addressed my concerns, and I will maintain my score.

---

### Official Review · Reviewer_ugkR · 2026-03-10

**Soundness:** 4
**Presentation:** 4
**Significance:** 4
**Originality:** 4
**Overall Recommendation:** 5
**Confidence:** 5

**Summary:**

This paper proposes UDPD, an uncertainty-modulated dual-path diffusion model for representation learning on text-attributed graphs (TAGs). The core motivation is to address two practical problems simultaneously: (1) the structural–semantic mismatch arising from divergent modality distributions of graph topology and node text, and (2) dual-source noise from both graph structure and textual content. The proposed framework consists of three main components: a dual-perspective encoder that separately produces semantic embeddings (via LLM-extracted keywords and Sentence-BERT) and structural embeddings (via GCN); a cooperative dual-path diffusion mechanism where the two branches mutually condition each other through a cross-attention strategy; and an uncertainty-modulated gating mechanism that uses per-node predictive entropy to adaptively regulate the interaction intensity between branches. Experiments are conducted on five citation graph benchmarks (Cora, Citeseer, PubMed, ArXiv-2023, and OGBN-ArXiv), and the method outperforms a range of GNN and LLM-based baselines on node classification.

**Compliance With Llm Reviewing Policy:**

Affirmed.

**Key Questions For Authors:**

1. What happens if instead of computing $h_i^{\text{rsd}} = h_i^{\text{sem}} - h_i^{\text{str}}$, you simply treat $h_i^{\text{sem}}$ directly as the input to the semantic branch? Is the residual formulation strictly necessary, or is this an implementation artifact?

2. Did you experiment with any alternatives to the $(1 + u_i)$ scaling factor, such as a learned sigmoid gate or a soft attention temperature? If not, why was this specific form chosen?

3. What are the wall-clock training and inference times on OGBN-ArXiv compared to TAPE and GraphBridge? How does the number of diffusion steps $T$ affect both performance and runtime?

**Limitations:**

Yes

**Strengths And Weaknesses:**

S1. Well-motivated problem formulation. The paper identifies a practically important and under-addressed challenge: existing TAG methods typically perform hard modality fusion directly on raw inputs, making them vulnerable to both structural–semantic mismatch and dual-source noise. The framing of this twin problem is clear and the motivation is substantiated by empirical examples (e.g., GAT collapsing to 53.50% on PubMed despite competitive performance elsewhere).

S2. While diffusion models have been successfully applied to graph generation tasks (e.g., GDSS at ICML 2022, DiGress at ICLR 2023), their use for representation learning on TAGs remains underexplored. UDPD makes a deliberate design choice to operate in a continuous latent space (thus circumventing the challenge of diffusing over discrete graph topologies), which is principled and well-motivated in Appendix C.1. The reinterpretation of inference as "feature purification" rather than standard generative sampling is elegant.

S3. Solid empirical performance. UDPD achieves consistent gains across all five benchmarks, including meaningful improvements over strong recent baselines such as GraphBridge and TAPE. The ablation study in Table 3 is well-designed and isolates the contribution of each major component, with results pointing to the dual-path interaction (rather than, say, LLM initialization alone) as the primary driver of performance.

W1. If the projections do not land in a shared meaningful space (e.g., they differ in scale, angle, or basis), the residual may carry noise from representational misalignment rather than genuine semantic deviation from neighborhood context. The paper should provide either a formal guarantee or ablation evidence (e.g., with/without the explicit residual vs. concatenation or other fusion alternatives) to justify this design.

W2. On OGBN-ArXiv (169K nodes), the per-epoch cost of running the dual-path diffusion and recomputing node uncertainty is non-trivial. Training for 500 epochs under this setup is computationally demanding, yet the paper reports no wall-clock times, GPU memory footprints, or convergence curves

---

> ### Author Rebuttal · Authors · 2026-03-31
>
> **W1&Q1**: Thanks for your insightful suggestion! Directly using $h_i^{\text{sem}}$ may cause the structural and semantic branches to learn overlapping signals that can be explained by either modality. The residual formulation explicitly encourages the semantic branch to focus on the semantic deviation from structural patterns, thereby promoting complementary representations and reducing redundancy.
>
> In addition, we have added an ablation study to evaluate the model’s performance with/without the explicit residual. As shown in Table 1, removing the explicit residual consistently leads to worse performance across all five datasets, confirming that the residual design is crucial rather than merely an implementation artifact.
>
> Table 1. Ablation Study of the Explicit Residual
>
> | Methods                    | Cora           | Citeseer       | PubMed         | ArXiv-2023     | OGBN-ArXiv     |
> | :------------------------ | :------------: | :------------: | :------------: | :------------: | :------------: |
> | Without Explicit Residual | 90.48±2.67     | 81.07±1.11     | 87.81±0.52     | 74.19±0.28     | 69.98±0.17     |
> | With Explicit Residual    | **92.73±0.58** | **85.89±1.12** | **94.93±0.50** | **83.34±0.92** | **78.55±1.31** |
>
>
>
> **W2&Q3**: Thanks! On a single NVIDIA RTX 3090 (24GB) GPU, the wall-clock training time of our UDPD on the OGBN-ArXiv dataset for 500 epochs is 1517.23 seconds (~25.2 minutes), with a peak GPU memory usage of 20.9 GB. In comparison, baselines such as TAPE and GraphBridge typically require 1-2 days to complete training, highlighting the computational efficiency of UDPD.
>
> We further conduct experiments to examine how varying the number of diffusion steps $T$ influences classification performance (Table 2) and inference runtime (Table 3). As shown, our UDPD achieves a favorable trade-off between performance and computational cost. Performance peaks at $T=10$ for small- and medium-sized graphs and at $T=15$ for large graphs, while inference runtime grows roughly linearly with $T$.
>
>
>
> Table 2. Performance with Different Diffusion Steps $T$ in Accuracy (%)
>
> | Datasets    | T=5        | T=10           | T=15           | T=20       |
> | :--------- | :--------: | :------------: | :------------: | :--------: |
> | Cora       | 90.41±2.34 | **92.73±0.58** | 89.58±2.17     | 89.48±1.30 |
> | Citeseer   | 82.04±3.06 | **85.89±1.12** | 82.17±2.22     | 81.61±2.64 |
> | PubMed     | 89.43±1.18 | **94.93±0.50** | 92.43±0.18     | 91.37±0.17 |
> | ArXiv-2023 | 77.04±0.28 | 81.68±0.44     | **83.34±0.92** | 82.13±0.29 |
> | OGBN-ArXiv | 72.29±0.20 | 75.28±0.95     | **78.55±1.31** | 76.44±3.91 |
>
> Table 3. Inference Time (seconds) on OGBN-ArXiv with Different $T$
>
> | Metrics             | T=5       | T=10      | T=15      | T=20      |
> | :----------------- | :-------: | :-------: | :-------: | :-------: |
> | Inference Time (s) | 257.4±8.3 | 493.8±7.5 | 794.6±2.1 | 965.6±2.6 |
>
> **Q2**: Following your suggestion, we have added a comparative experiment to evaluate the $(1+u_i)$ scaling factor against a learned sigmoid gate. As shown in Table 4, compared with the $(1+u_i)$ scaling factor, the learned sigmoid gate consistently yields inferior performance across all datasets, indicating that the proposed uncertainty-aware scaling mechanism is more effective in facilitating cross-branch information interaction.
>
> Table 4. Performance Comparison of Different Scaling Factors
>
> | Methods               | Cora           | Citeseer       | PubMed         | ArXiv-2023     | OGBN-ArXiv     |
> | :------------------- | :------------: | :------------: | :------------: | :------------: | :------------: |
> | Learned Sigmoid Gate | 90.90±1.48     | 81.82±2.71     | 88.50±0.18     | 79.08±0.38     | 75.97±0.12     |
> | $(1+u_i)$            | **92.73±0.58** | **85.89±1.12** | **94.93±0.50** | **83.34±0.92** | **78.55±1.31** |

---

> > ### Author Rebuttal · Reviewer_ugkR · 2026-04-03
> >
> > thanks for the response and I will maintain my score.

---

### Official Review · Reviewer_KtBN · 2026-03-12

**Soundness:** 3
**Presentation:** 2
**Significance:** 3
**Originality:** 3
**Overall Recommendation:** 4
**Confidence:** 4

**Summary:**

The paper presents a representation learning method for text-attributed graphs (TAGs). The task is challenging due to the mismatch between structural-semantic modalities due to their divergent distributions, and dual-source noise inherent in node textual content and graph structure. Existing approaches employ a rigid fusion of the two modalities, but overlook their inherent noise, which may lead to the amplification of such noises during information propagation. As such, the authors propose UDPD, which (1) employ a dual-perspective node encoding to learn semantic and structural embeddings; (2) incorporate a diffusion framework on both semantic and structural branches, with mutual interaction between the two branches. Experiments are conducted on five public benchmarks to validate the effectiveness of UDPD.

**Compliance With Llm Reviewing Policy:**

Affirmed.

**Final Justification:**

The authors largely answered my concerns during rebuttal. Therefore, I raised my score to weak accept, although I think W3 remains a important concern. Perhaps some empirical experiments could be designed to validate the authors' argument.  They are strongly encouraged to incorporate the suggested revisions in future versions.

**Key Questions For Authors:**

Please see weaknesses above.

Additionally, the performance of GPT-4o is often significantly worse than much smaller models like BERT in table 2. This is counter-intuitive. More analysis and explanation is needed.

**Limitations:**

The author mainly discussed the limitations of prior method, but not much on their own method. The authors could consider discuss what kind of semantic-structural mismatch their method could handle or is best suited to.

**Strengths And Weaknesses:**

Strengths:

S1. The paper addresses an important problem in TAGs, which aims to integrate structural-semantic effectively. This is an important and timely topics.

S2. The use of diffusion model is considered novel in the modeling of TAGs.

S3. Experiments show promising results across 5 benchmark datasets.

Weaknesses:

W1. The motivation of the paper can be strengthen. The claim of structure-semantic mismatch could be demonstrated better using examples and statistics from the actual data. There could also be different kinds of mismatch (e.g., different topics; the same topic but different aspects; direct contradiction or inconsistencies; different but complementary). Additionally, how the diffusion model can address such mismatch, is not well explained.

W2. The structural/semantic denoising network is quite opaque. What is its key mechanism? How is the cross-attention used in the denoising networks?

W3. How does the alternating update strategy mitigate the amplification of noises in the two branches? The interaction could potentially worsen the amplification, if not designed properly.

W4. Experimental results mainly focus on performance. How about some case studies showing the denoising effect? Some quantitative results on the extend of denoising could also shed more insights.

---

> ### Author Rebuttal · Authors · 2026-03-31
>
> **W1**: Thanks! The structure-semantic mismatch naturally exists in actual data. For example, a CV paper may cite an NLP paper to adopt a mathematical formulation, linking nodes discussing different topics, while two papers may cite each other to present opposing viewpoints, leading to semantic contradictions.
>
> We further provide quantitative statistics to demonstrate the structure-semantic mismatch. Specifically, we measure the semantic similarity between neighboring nodes, as well as the distributional divergence between structural neighborhoods and semantic clusters. As shown in Table 1, neighboring nodes exhibit relatively low semantic similarity, while the Jensen-Shannon Divergence (JSD) values between structural neighborhoods and semantic clusters consistently exceed 0.4, indicating a significant distributional discrepancy. These results indicate that structure and semantics are often misaligned in real-world text-attributed graphs.
>
> Table 1. Statistics of Structure-Semantic Mismatch on Actual Data
>
> | Metrics              | Cora  | Citeseer | PubMed | ArXiv-2023 | OGBN-ArXiv |
> | :------------------ | :---: | :------: | :----: | :--------: | :--------: |
> | Semantic Similarity | 0.480 | 0.563    | 0.356  | 0.546      | 0.454      |
> | JSD                 | 0.511 | 0.509    | 0.727  | 0.557      | 0.419      |
>
> Our UDPD addresses structure–semantic mismatch via an uncertainty-modulated dual-path diffusion. The forward process gradually injects noise into structural and semantic embeddings, and the reverse process denoises them jointly. During denoising, the two branches interact to suppress inconsistent information while preserving complementary signals, progressively refining node embeddings and mitigating structure–semantic discrepancies.
>
> **W2**: The key mechanism of our structural/semantic denoising network is a Multilayer Perceptrons (MLPs)-based architecture augmented with cross-attention, enabling interactive refinement between the structural and semantic branches. Taking the structural denoiser as an example, the network treats the noisy structural feature as the "Query", and the cleaner semantic feature from the parallel branch as both the "Key" and "Value". This design allows the structural branch to selectively extract reliable semantic signals to suppress its own noise, making the denoising process interpretable.
>
> **W3**: We fully agree that, if not designed properly, the interaction between the two branches could potentially worsen the amplification. To prevent this, we adopt an iterative alternating update strategy. At each reverse step, we first update the structural branch, and then use its newly refined, cleaner state as a reliable reference to guide the semantic branch. By updating sequentially in this manner, each branch consistently provides a cleaner reference for the other, preventing error propagation and effectively mitigating noise amplification.
>
> **W4**: Following your suggestion, we have added quantitative experiments to evaluate the denoising effect by comparing the semantic similarity between neighboring nodes before and after denoising. As shown in Table 2, our method significantly enhances the node similarity after denoising, effectively enhancing the alignment between structural and semantic embeddings.
>
> Table 2. Semantic Similarity between Neighboring Nodes
>
> | States  | Cora      | Citeseer  | PubMed    | ArXiv-2023 | OGBN-ArXiv |
> | :----- | :-------: | :-------: | :-------: | :--------: | :--------: |
> | Before | 0.480     | 0.563     | 0.356     | 0.546      | 0.454      |
> | After  | **0.672** | **0.851** | **0.552** | **0.793** | **0.680** |
>
> Also of note, in Section 4.4 in our ICML submission, we present a t-SNE visualization that illustrates the global denoising effect. As shown, after denoising, nodes of the same class form clear clusters, providing an intuitive demonstration that dual-source noise has been effectively removed.
>
> **Q1**: Thanks! The performance difference is mainly due to the number of dataset classes and the distinct architectures of models. BERT employs supervised fine-tuning with a dedicated classification head. On datasets with only a few classes (i.e., Cora and PubMed), this head can be easily trained to fit the labels. In contrast, GPT-4o lacks a task-specific classification head and relies on its broad internal knowledge for generative reasoning. On simpler tasks, it may “over-think” and struggle with strict label assignments. However, this trend reverses as the number of classes increases. For example, on the ArXiv-2023 dataset with 40 classes, GPT-4o achieves 64.86%, significantly outperforming BERT at 58.97%. This demonstrates that GPT-4o excels in more complex classification spaces, where its reasoning capabilities become advantageous.

---

> > ### Author Rebuttal · Reviewer_KtBN · 2026-04-04
> >
> > I appreciate authors' response, which clarified some of my questions. Some follow ups:
> >
> > W1 - while the quantitative stats are useful, can authors provide some deeper insights, such as different types of semantic-structural mismatch?
> >
> > W3 - The response is not fully convincing. If in any step it actually gets worse, it will get into a downward spiral

---

> > > ### Author Response · Authors · 2026-04-07
> > >
> > > Thank you again for your insightful comments! The responses to your questions are listed below:
> > >
> > > **W1**: To provide deeper insights, we categorize the structural-semantic mismatches observed in our datasets into four main types:
> > >
> > > 1\) **Different topics**. In broad datasets such as OGBN-ArXiv and ArXiv-2023, cross-field connections frequently occur. As noted in our previous response, a computer vision paper may cite an NLP paper to to reuse a model or methodology. Although a structural edge exists, their textual semantics may differ substantially. Quantitatively, this type accounts for 26.86% of connected edges in OGBN-ArXiv and 24.40% in ArXiv-2023.
> > >
> > > 2\) **The same topic but different aspects**. In datasets like Cora and Citeseer, connected nodes often share a high-level research area but differ significantly in their specific focus. For instance, two papers in Cora may both belong to "Reinforcement Learning" , yet one focuses on theoretical analysis while the other studies robotic control applications. This type accounts for 23.59% of connected edges in Cora and 19.57% in Citeseer.
> > >
> > > 3\) **Semantic mismatch in specialized domains**. In domain-specific datasets like PubMed, papers often rely on tools, evidence, or findings from different subtopics. As a result, structurally connected papers may still exhibit substantial semantic differences. In PubMed, 35.17% of connected edges fall into the different topics category, while 39.90% correspond to the same topic but different aspects category, indicating that semantic mismatch is also prevalent in specialized domains.
> > >
> > > 4\) **Direct contradiction or inconsistencies**. Across all datasets, a structural edge does not necessarily imply agreement between two papers. For instance, a recent work may cite a previous method primarily to highlight its limitations. If a graph model directly aggregates features from such semantically inconsistent nodes, it may introduce harmful noise. Our analysis using Natural Language Inference (NLI) to detect textual contradictions further supports this observation, identifying potentially contradictory or inconsistent edges in 26.17% of connected pairs in Cora, 23.98% in Citeseer, 39.10% in PubMed, 33.60% in OGBN-ArXiv, and 33.92% in ArXiv-2023.
> > >
> > > These four types highlight the complex interplay between graph structure and textual semantics in real-world graphs. Our proposed method is designed to effectively address such structural-semantic mismatches.
> > >
> > > **W3**: Our method is designed to prevent such a "downward spiral" through two mechanisms. First, the interaction between the structural and semantic branches is mediated by a cross-attention mechanism, which functions as a feature filter rather than direct feature mixing. When one branch produces noisy or conflicting signals at a certain step, the attention mechanism naturally assigns lower weights to these unreliable features, limiting their influence on the other branch. In addition, the clean initial embeddings are retained as a stable anchor, ensuring that information exchange remains grounded even during the noisy early stages.
> > >
> > > Second, the diffusion process follows a pre-defined small-variance noise schedule, meaning that any error introduced at a particular step is strictly limited in magnitude. More importantly, the entire denoising process is supervised by the end-to-end objective of reconstructing the original clean representations, which provides a global optimization signal and prevents local errors from accumulating across steps.
> > >
> > > Together, these mechanisms ensure that errors do not propagate uncontrollably during the diffusion process.

---

### Official Review · Reviewer_xrWt · 2026-03-13

**Soundness:** 2
**Presentation:** 2
**Significance:** 2
**Originality:** 2
**Overall Recommendation:** 2
**Confidence:** 2

**Summary:**

The paper analyzes a general context where graph topology and node semantics are often misaligned or noisy, which may degrade the performance of standard graph neural networks that rely on message passing. The authors claim to investigate the concept of structural–semantic mismatch and propose a new framework called Uncertainty-Modulated Dual-Path Diffusion (UDPD) to address this problem.

**Compliance With Llm Reviewing Policy:**

Affirmed.

**Final Justification:**

I have reviewed all reviewer comments and the authors’ rebuttal. I appreciate the detailed responses, and some of my concerns have been addressed. However, I still have reservations about some conclusions and results. I will maintain my rating but lower my confidence.

**Key Questions For Authors:**

- The paper argues that many text-attributed graph (TAG) datasets suffer from structural–semantic mismatch, where graph topology and textual attributes follow different distributions. It would be helpful to report quantitative statistics such as:
  - semantic similarity between neighboring nodes versus randomly sampled nodes,
  - mutual information between topology and textual embeddings,
  - distributional divergence between structural neighborhoods and semantic clusters.
- Text-attributed graphs involve discrete graph topology and sparse, high-dimensional textual attributes. How diffusion is adapted to graph data in which features are discrete:
  - How the diffusion process is defined over graph topology (e.g., edges or adjacency matrices).
  - How semantic attributes are corrupted and denoised.
  - Whether the diffusion process preserves important graph invariants, such as connectivity patterns or degree distributions.
  - How the proposed dual-path diffusion avoids introducing unrealistic structures during the denoising process.
- The paper does not clearly define how node uncertainty is computed. Is it derived from prediction entropy, feature variance, or another metric?
- In Section 3.2, the semantic embedding module leverages LLM-based filtering, while the structural embedding branch appears to rely on raw textual features. Is there any reason for adopting this special design?
- The paper does not clearly report the search space for hyperparameters.
- The proposed framework integrates several components, including GNN backbone, diffusion-based representation refinement, LLM-based semantic embeddings. However, the paper does not provide ablation studies examining how sensitive the method is to the choice of these components.

**Limitations:**

yes

**Strengths And Weaknesses:**

**Strengths**
- Working on an important challenge in text-attributed graph learning;
- The idea of using node uncertainty to dynamically control cross-modal interaction between structural and semantic branches is an interesting design choice.

**Weaknesses**
- Limited clarity on how diffusion is adapted to discrete graph structures;
- Experimental analysis lacks deeper ablation and robustness studies.

---

> ### Author Rebuttal · Authors · 2026-03-31
>
> **Q1**: Thanks for your suggestions! 1) We first measured cosine similarity between textual embeddings of neighboring and randomly sampled nodes. Across datasets, neighboring nodes exhibit higher semantic similarity than randomly ones (i.e., 0.480 vs. 0.189, 0.563 vs. 0.176, 0.356 vs. 0.131, 0.546 vs. 0.175, and 0.454 vs. 0.357 on Cora, Citeseer, PubMed, ArXiv-2023, and OGBN-ArXiv, respectively). Nevertheless, the moderate similarity of neighboring nodes demonstrates that graph structure only partially reflects semantic similarity, confirming structural–semantic mismatch.
>
> 2\) We then estimated the mutual information between topology and textual embeddings, and the low values (i.e., 0.0080/0.0039/0.0150/0.0020/0.0400 on Cora/Citeseer/PubMed/ArXiv-2023/OGBN-ArXiv, respectively) indicate that topology provides limited information about node semantics, highlighting structural–semantic mismatch.
>
> 3\) We finally calculated the distributional divergence between structural neighborhoods and semantic clusters using Jensen-Shannon Divergence (JSD). The average JSD is high (i.e., 0.511/0.509/0.727/0.557/0.419 on Cora/Citeseer/PubMed/ArXiv-2023/OGBN-ArXiv, respectively), indicating that structural neighborhoods differ substantially from semantic clusters, further verifying structural–semantic mismatch.
>
> **W1&Q2**: Since diffusion operates on continuous variables, our UDPD map the graph topology and textual attributes into a continuous latent space using neural encoders.
>
> 1\) Instead of applying diffusion directly to edges or the adjacency matrix, we encode graph topology with a GCN, which aggregates neighborhood information into continuous node embeddings, and perform diffusion over these embeddings.
>
> 2\) For semantic attributes, we first encode them using Sentence-BERT. The forward process corrupts these continuous embeddings by progressively adding standard Gaussian noise, while the reverse process employs MLPs with cross-attention to denoise and recover the clean semantic features.
>
> 3\) Yes. Since diffusion is performed solely on continuous node embeddings, the original graph topology remains intact, preserving important graph invariants like connectivity patterns or degree distributions.
>
> 4\) Unlike diffusion methods that reconstruct graph topology, our UDPD reconstructs node embeddings. The diffusion process is performed on continuous embeddings rather than discrete edges or adjacency matrices, preventing unrealistic structures. Moreover, during denoising, structural embeddings are constrained by purified semantic embeddings via cross-attention.
>
> **Q3**: Node uncertainty is computed using normalized predictive entropy derived from the model’s output probability distribution. The exact formulation is given in Equations (16) and (17) in Section 3.3, with further theoretical justification provided in Appendix D.
>
> **Q4**: This design reflects the different information needs of the semantic and structural branches. The semantic branch uses LLM-based filtering to extract core keywords and remove noise, producing clean node features. The structural branch encodes raw text to preserve full contextual information, which is essential for capturing neighborhood and topological patterns.
>
> **Q5**: Thanks! The hyperparameter search space is as follows: noise schedule $\beta \in \lbrace 10^{-4}, 10^{-3}, \dots, 0.02 \rbrace$, diffusion steps $T \in \lbrace 1, 5, 10, 15, 20 \rbrace$, top-$K$ keywords $\in \lbrace 5, 10, 20 \rbrace$, loss weights $\lambda_1, \lambda_2 \in \lbrace 0.4, 0.5, 0.6 \rbrace$, and hidden dimensions $\in \lbrace 128, 256, 512 \rbrace$.
>
> **W2&Q6**: Thanks! We have added ablation experiments replacing the GCN backbone with GAT and GraphSAGE. As shown in Table 1, our UDPD maintains high performance across different GNN backbones, demonstrating its robustness.
>
> Table 1. Effect of GNN Backbones on Accuracy (%)
>
> | Backbones  | Cora           | CiteSeer       | PubMed         | ArXiv-2023     | OGBN-ArXiv     |
> | :-------- | :------------: | :------------: | :------------: | :------------: | :------------: |
> | GCN       | **92.73 ± 0.58** | **85.89 ± 1.12** | 94.93 ± 0.50     | 83.34 ± 0.92     | **78.55 ± 1.30** |
> | GAT       | 91.25 ± 1.00     | 84.01 ± 0.79     | **95.07 ± 0.11** | **83.92 ± 1.77** | 77.39 ± 1.50     |
> | GraphSAGE | 92.06 ± 0.22     | 84.76 ± 2.03     | 94.51 ± 1.30     | 81.65 ± 2.38     | 77.02 ± 0.99     |
>
> For ablation study on LLM-based semantic embeddings, please refer to Q2 of Reviewer 4. Also of note, for ablation study on diffusion-based representation refinement, we evaluated the model without the semantic (w/o Sem-Diff) and structural (w/o Str-Diff) diffusion branches, as reported in Section 4.3. Both variants show a performance drop, confirming the contribution of each branch and the effectiveness of the dual-path design.

---

> > ### Author Rebuttal · Reviewer_xrWt · 2026-04-03
> >
> > Thanks for authors' clarifications. While most of my concerns are addressed, I remain unconvinced about two aspects:
> > - The use of LLM-based filtering introduces an additional source of improvement that is not applied to baselines, making it difficult to disentangle the contribution of the proposed diffusion framework from enhanced feature engineering.
> > - Thanks for clarification that the diffusion process is performed on continuous embeddings. But the method subtracts structural and semantic embeddings produced by different encoders, implicitly assuming a shared latent space; however, aligning dimensions via projection does not guarantee meaningful alignment of representations.

---

> > > ### Author Response · Authors · 2026-04-04
> > >
> > > Thank you again for your insightful comments! The responses to your questions are listed below:
> > >
> > > **Q1**: Following your suggestion, we have added additional experiments by evaluating standard baselines (e.g., GCN and GAT) using both raw text and LLM-extracted features. As shown in Table 1, replacing raw text with LLM-extracted features degrades the performance of baselines across all datasets, indicating that LLM-extracted features do not automatically improve model performance. A possible reason is that such features emphasize global semantic cues while discarding contextual information, which may reduce feature similarity among neighboring nodes and introduce noise during neighborhood aggregation.
> > >
> > > Also of note, Table 3 in our ICML submission presents the ablation results of our model without LLM-extracted features. Despite using the same raw text features as the baselines, our model still achieves competitive performance across datasets (i.e., 89.58%, 84.04%, 88.56%, 81.43%, and 71.21% on Cora, Citeseer, PubMed, ArXiv-2023, and OGBN-ArXiv, respectively), suggesting that the observed improvements are mainly attributed to the proposed diffusion framework rather than enhanced feature engineering.
> > >
> > > Table 1. Performance comparison of baselines using raw text versus LLM-extracted features.
> > >
> > > | **Methods**        | **Cora**       | **Citeseer**   | **PubMed**     | **ArXiv-2023** | **OGBN-ArXiv** |
> > > | ------------------ | -------------- | -------------- | -------------- | -------------- | -------------- |
> > > | GCN (Raw Text)     | **84.97±0.34** | **71.97±0.33** | **50.11±2.04** | **74.38±0.33** | **70.76±0.16** |
> > > | GCN (LLM Features) | 81.93±0.16     | 69.43±0.63     | 47.48±1.25     | 69.37±0.34     | 67.23±0.22     |
> > > | GAT (Raw Text)     | **90.39±0.46** | **72.74±0.42** | **53.50±1.15** | **70.46±0.20** | **68.63±0.11** |
> > > | GAT (LLM Features) | 89.16±0.31     | 69.21±0.85     | 50.21±2.24     | 65.21±0.39     | 64.62±0.25     |
> > >
> > >
> > >
> > >
> > >
> > > **Q2:** There seems to be a misunderstanding regarding how alignment between embeddings is achieved in our method. Following prior work on cross-modal representation learning, such as CLIP [Radford et al. ICML 2021] and Graphformers [Yang et al. NeurIPS 2021], we project the structural and semantic embeddings produced by different encoders into a shared latent space. This projection mainly serves as an initialization step that enables subsequent operations and interactions between the two embeddings. The meaningful alignment is achieved through end-to-end joint optimization. In particular, we introduce a cross-attention interaction mechanism during the denoising process, allowing the structural and semantic embeddings to exchange information and progressively refine each other under a unified training objective.
> > >
> > > Also of note, Table 3 in our ICML submission provides empirical evidence supporting this point. When removing the cross-attention interaction module (w/o Interaction), the performance drops noticeably across all datasets. If the projection alone were sufficient to align the structural and semantic embeddings, removing the interaction module would have little impact. The observed degradation therefore indicates that projection only enables the representations to share a space, while meaningful alignment is achieved through the interaction process and end-to-end training.

---

### Decision · Program_Chairs · 2026-04-30

**Decision:**

Accept (regular)

**Comment:**

The paper analyzes a general context where graph topology and node semantics are often misaligned or noisy, which may degrade the performance of standard graph neural networks that rely on message passing. The paper is in a good quality, and all issues are addressed. I would recommend it as accepted.